


# Climate seasonality limits carbon assimilation and storage in tropical forests

Fabien H. Wagner[1], Bruno Hérault[2], Damien Bonal[3], Clément Stahl[4,5], Liana O. Anderson[6], Timothy R. Baker[7], Gabriel Sebastian Becker[8], Hans Beeckman[9], Danilo Boanerges Souza[10], Paulo Cesar Botosso[11], David M.J.S. Bowman[12], Achim Bräuning[13], Benjamin Brede[14], Foster Irving Brown[15], Jesus Julio Camarero[16,17], Plínio Barbosa Camargo[18], Fernanda C.G. Cardoso[19], Fabrício Alvim Carvalho[20], Wendeson Castro[21], Rubens Koloski Chagas[22], Jérome Chave[23], Emmanuel N. Chidumayo[24], Deborah A. Clark[25], Flavia Regina Capellotto Costa[26], Camille Couralet[9], Paulo Henrique da Silva Mauricio[15], Helmut Dalitz[8], Vinicius Resende de Castro[27], Jaçanan Eloisa de Freitas Milani[28], Edilson Consuelo de Oliveira[29], Luciano de Souza Arruda[30], Jean-Louis Devineau[31], David M. Drew[32], Oliver Dünisch[33], Giselda Durigan[34], Elisha Elifuraha[35], Marcio Fedele[36], Ligia Ferreira Fedele[36], Afonso Figueiredo Filho[37], César Augusto Guimarães Finger[38], Augusto César Franco[39], João Lima Freitas Júnior[21], Franklin Galvão[28], Aster Gebrekirstos[40], Robert Gliniars[8], Paulo Maurício Lima de Alencastro Graça[41], Anthony D. Griffiths[42,43], James Grogan[44], Kaiyu Guan[45,46], Jürgen Homeier[47], Maria Raquel Kanieski[48], Lip Khoon Kho[49], Jennifer Koenig[43], Sintia Valerio Kohler[37], Julia Krepkowski[13], José Pires Lemos-Filho[50], Diana Lieberman[51], Milton Eugene Lieberman[51], Claudio Sergio Lisi[36,52], Tomaz Longhi Santos[28], José Luis López Ayala[53], Eduardo Eijji Maeda[54], Yadvinder Malhi[55], Vivian R.B. Maria[36], Marcia C.M. Marques[19], Renato Marques[56], Hector Maza Chamba[57], Lawrence Mbwambo[58], Karina Liana Lisboa Melgaço[26], Hooz Angela Mendivelso[16,17], Brett P. Murphy[59], Joseph J. O'Brien[60], Steven F. Oberbauer[61], Naoki Okada[62], Raphaël Pélissier[63,64], Lynda D. Prior[12], Fidel Alejandro Roig[65], Michael Ross[66], Davi Rodrigo Rossatto[67], Vivien Rossi[68], Lucy Rowland[69], Ervan Rutishauser[70], Hellen Santana[26], Mark Schulze[71], Diogo Selhorst[72], Williamar Rodrigues Silva[73], Marcos Silveira[15], Susanne Spannl[13], Michael D. Swaine[74], José Julio Toledo[75], Marcos Miranda Toledo[76], Marisol Toledo[77], Takeshi Toma[78], Mario Tomazello Filho[36], Juan Ignacio Valdez Hernández[53], Jan Verbesselt[14], Simone Aparecida Vieira[79], Grégoire Vincent[64], Carolina Volkmer de Castilho[80], Franziska Volland[13], Martin Worbes[81], Magda Lea Bolzan Zanon[82], and Luiz E.O.C. Aragão[1]

[1]Remote Sensing Division, National Institute for Space Research - INPE, São José dos Campos 12227-010, SP, Brazil
[2]CIRAD, UMR Ecologie des Forêts de Guyane, Kourou 97379, France
[3]INRA, UMR EEF 1137, Champenoux 54280, France
[4]INRA, UMR Ecologie des Forêts de Guyane, Kourou 97387, France
[5]Department of Biology, University of Antwerp, Wilrijk 2610, Belgium
[6]National Center for Monitoring and Early Warning of Natural Disasters - CEMADEN, São José dos Campos 12.247-016, SP, Brazil
[7]School of Geography, University of Leeds, Leeds LS2 9JT, UK
[8]Institute of Botany, University of Hohenheim, Stuttgart 70593, Germany
[9]Laboratory for Wood Biology and Xylarium, Royal Museum for Central Africa, Tervuren B-3080, Belgium
[10]Programa de Pós-graduação em Ciências de Florestas Tropicais, Instituto Nacional de Pesquisas da Amazônia, Manaus 69067-375, AM, Brazil
[11]Embrapa Florestas, Brazilian Agricultural Research Corporation, Colombo 83411-000, PR, Brazil



[12]School of Biological Sciences, University of Tasmania, Hobart 7001, Tasmania, Australia

[13]Institute of Geography, University of Erlangen-Nuremberg, Erlangen 91058, Germany

[14]Laboratory of Geo-information Science and Remote Sensing, Wageningen University, Wageningen 6708PB, The Netherlands

[15]Centro de Ciências Biológicas e da Natureza, Laboratório de Botânica e Ecologia Vegetal, Universidade Federal Do Acre, Rio Branco 69915-559, AC, Brazil

[16]Instituto Pirenaico de Ecologia, Consejo Superior de Investigaciones Cientificas (IPE-CSIC), Zaragoza 50059, Spain

[17]Instituto Boliviano de Investigacion Forestal (IBIF), Santa Cruz de la Sierra 6204, Bolivia

[18]Centro de Energia Nuclear na Agricultura, Laboratório de Ecologia Isotópica, Universidade de São Paulo, Piracicaba 13416903, SP, Brazil

[19]Departamento de Botânica, Universidade Federal do Paraná, Curitiba 81531-980, PR, Brazil

[20]Departamento de Botânica, Universidade Federal de Juiz de Fora (UFJF), Juiz de Fora 36015-260, MG, Brazil

[21]Programa de Pós-Graduação Ecologia e Manejo de Recursos Naturais, Universidade Federal do Acre, Rio Branco 69915-559, AC, Brazil

[22]Departamento de Ecologia do Instituto de Biociências, Universidade de São Paulo (USP), São Paulo 05508-090, SP, Brazil

[23]UMR 5174 Laboratoire Evolution et Diversité Biologique, CNRS & Université Paul Sabatier, Toulouse 31062, France

[24]Biological Sciences Department, University of Zambia, Lusaka Box 32379, Zambia

[25]Department of Biology, University of Missouri-St. Louis, Saint Louis 63121, MO, USA

[26]Coordenação de Pesquisas em Biodiversidade, Instituto Nacional de Pesquisas da Amazônia, Manaus 69080-971, AM, Brazil

[27]Departamento de Engenharia Florestal, Universidade federal de Viçosa (UFV), Viçosa 36570-000, MG, Brazil

[28]Departamento de Engenharia Florestal, Universidade Federal do Paraná, Curitiba 80210-170, PR, Brazil

[29]Centro de Ciências Biológicas e da Natureza, Laboratório de Botânica e Ecologia Vegetal, Universidade Federal do Acre, Rio Branco 69915-559, AC, Brazil

[30]Prefeitura Municipal de Rio Branco, Rio Branco 69900-901, AC, Brazil

[31]Département Hommes, Natures, Sociétés, Centre National de la Recherche Scientifique (CNRS) et UMR 208 Patrimoines Locaux et Gouvernance, Paris 75231 cedex 05, France

[32]Dept. Forest and Wood Science, University of Stellenbosch, Stellenbosch 7600, South Africa

[33]Meisterschule Ebern für das Schreinerhandwerk, Ebern 96106, Germany

[34]Floresta Estadual de Assis, Assis 19802-970, SP, Brazil

[35]Tanzania Forestry Research Institute (TAFORI), Dodoma P. O. Box 1576, Tanzania

[36]Departamento de Ciências Florestais, Universidade de São Paulo, Escola Superior de Agricultura Luiz de Queiroz, Piracicaba 13418-900, SP, Brazil

[37]Departamento de Engenharia Florestal - DEF, Universidade Estadual do Centro-Oeste, Irati 84500-000, PR, Brazil

[38]Departamento de Ciências Florestais, Centro de Ciências Rurais, Universidade Federal de Santa Maria, Santa Maria 97105-9000, RS, Brazil

[39]Departamento de Botânica, Laboratório de Fisiologia Vegetal, Universidade de Brasília, Instituto de Ciências Biológicas, Brasília 70904-970, DF, Brazil

[40]World Agroforestry Centre (ICRAF), Nairobi PO Box 30677-00100, Kenya

[41]Coordenação de Pesquisa em Ecologia, Instituto Nacional de Pesquisas da Amazônia, Manaus C.P. 478 69011-970, AM, Brazil

[42]Departement of Land Resource Management, Northern Territory Government, Palmerston NT 0831 , Australia

[43]Research Institute for Environment and Livelihoods, Charles Darwin University, Darwin NT 0909, Australia

[44]Department of Biological Sciences, Mount Holyoke College, South Hadley 01075, MA, USA

[45]Department of Earth System Science, Stanford University, Stanford 94305, CA, USA

[46]Department of Natural Resources and Environmental Sciences, University of Illinois at Urbana Champaign, Champaign 61801, USA

[47]Department of Plant Ecology, Albrecht von Haller Institute of Plant Sciences, University of Göttingen, Göttingen 37073, Germany





[48]Departamento de Engenharia Florestal, Universidade do Estado de Santa Catarina - UDESC, Lages 88520-000, SC, Brazil

[49]Tropical Peat Research Institute, Biological Research Division, Malaysian Palm Oil Board, Selangor 43000, Malaysia

[50]Departamento de Botânica, Instituto de Ciências Biologicas, Universidade Federal de Minas Gerais, Belo Horizonte 31270-901, MG, Brazil

[51]Division of Science & Environmental Policy, California State University Monterey Bay, Seaside 93955, CA, USA

[52]Departamento de Biologia, Universidade Federal de Sergipe, São Cristóvão 49100-000, Brazil

[53]Programa Forestal, Colegio de Postgraduados, Montecillo 56230, México

[54]Department of Geosciences and Geography, University of Helsinki, Helsinki FI-00014, Finland

[55]School of Geography and the Environment, University of Oxford, Oxford OX1 3QY, UK

[56]Departamento de Solos e Engenharia Agrícola, Universidade Federal do Paraná, Curitiba 80035-050, PR, Brazil

[57]Laboratoria de Dendrochronologia y Anatomia de MaderasEspinoza, Universidad Nacional de Loja, Loja EC110103, Ecuador

[58]Tanzania Forestry Research Institute (TAFORI), Morogoro P. O. Box 1854, Tanzania

[59]Research Institute for the Environment and Livelihoods, Charles Darwin University, Darwin NT 0909, Australia

[60]Center for Forest Disturbance Science, USDA Forest Service, Athens 30607, GA, USA

[61]Department of Biological Sciences, Florida International University, Miami 33199, FL, USA

[62]Graduate School of Agriculture, Kyoto University, Kyoto 606-8501, Japan

[63]Institut Français de Pondicherry, Puducherry 6005001, India

[64]UMR AMAP (botAnique et bioinforMatique de l'Architecture des Plantes), IRD, Montpellier 34398, France

[65]Tree Ring and Environmental History Laboratory, Instituto Argentino de Nivología, Glaciología y Ciencias Ambientales - CONICET, Mendoza 5500, Argentina

[66]Department of Earth and Environment, Southeast Environmental Research Center, Florida International University, Miami 33199, FL, USA

[67]Departamento de Biologia Aplicada, FCAV, Universidade Estadual Paulista, UNESP, Jaboticabal 14884-000, SP, Brazil

[68]UR B&SEF (Biens et services des écosystèmes forestiers tropicaux), CIRAD, Yaoundé BP 2572, Cameroon

[69]School of Geosciences, University of Edinburgh, Edinburgh EH9 3FF, UK

[70]CarboForExpert (carboforexpert.ch), Geneva 1211, Switzerland

[71]HJ Andrews Experimental Forest, Oregon State University, Blue River 97413, OR, USA

[72]Ibama, Rio Branco 69907-150, AC, Brazil

[73]PRONAT - Programa de Pos-Graduação em Recurso Naturais, Universidade Federal de Roraima - UFRR, Boa Vista 69310-000, RR, Brazil

[74]School of Biological Sciences, University of Aberdeen, Aberdeen AB24 2TZ, UK

[75]Departamento de Ciências Ambientais, Universidade Federal do Amapá, Macapá 68902-280, AP, Brazil

[76]Embrapa Cocais, Brazilian Agricultural Research Corporation, São Luiz 65066-190, MA, Brazil

[77]Instituto Boliviano de Investigacion Forestal (IBIF), Universidad Autonoma Gabriel René Moreno, Santa Cruz de la Sierra CP 6201, Bolivia

[78]Department of Forest Vegetation, Forestry and Forest Products Research Institute (FFPRI), Ibaraki 305-8687, Japan

[79]Núcleo de Estudos e Pesquisas Ambientais (NEPAM), Universidade Estadual de Campinas (UNICAMP), Campinas 13083-867, SP, Brazil

[80]Embrapa Roraima, Brazilian Agricultural Research Corporation, Boa Vista 69301-970, RR, Brazil

[81]Crop Production Systems in the Tropics, Georg-August-University, Göttingen D-37077, Germany

[82]Departamento de Engenharia Florestal, Centro de Educação Superior Norte, Universidade Federal de Santa Maria, Frederico Westphalen 98400-000, RS, Brazil

*Correspondence to:* Fabien Hubert Wagner (wagner.h.fabien@gmail.com)





**Abstract.** The seasonal climate drivers of the carbon cycle in tropical forests remain poorly known, although these forests account for more carbon assimilation and storage than any other terrestrial ecosystem. Based on a unique combination of seasonal pan-tropical data sets from 89 experimental sites (68 include aboveground wood productivity measurements and 35 litter productivity measurements), their associate canopy photosynthetic capacity (enhanced vegetation index, EVI) and
5   climate, we ask how carbon assimilation and aboveground allocation are related to climate seasonality in tropical forests and how they interact in the seasonal carbon cycle. We found that canopy photosynthetic capacity seasonality responds positively to precipitation when rainfall is $< 2000$ mm.yr$^{-1}$ (water-limited forests) and to radiation otherwise (light-limited forests); on the other hand, independent of climate limitations, wood productivity and litterfall are driven by seasonal variation in precipitation and evapotranspiration respectively. Consequently, light-limited forests present an asynchronism between canopy
10   photosynthetic capacity and wood productivity. Precipitation first-order control indicates an overall decrease in tropical forest productivity in a drier climate.





## 1 Introduction

Tropical forests have a primary role in the terrestrial carbon (C) cycle, constituting 54% of the total aboveground biomass carbon of Earth's forests (Liu et al., 2015) and accounting for half ($1.19 \pm 0.41$ PgC yr$^{-1}$) of the global carbon sink of established forests (Pan et al., 2011; Baccini et al., 2012). While tropical forests have been acting as a long-term, net carbon sink,

a declining trend in carbon accumulation has been recently demonstrated for Amazonia (Brienen et al., 2015). Furthermore, a positive change in water-use efficiency of tropical trees due to the $CO_2$ increase has also been observed (van der Sleen et al., 2015). Understanding the seasonal drivers of the carbon cycle is needed to assess the mechanisms driving changes in forest carbon use and predict tropical forest behaviour under future climate changes.

Despite long-term investigation of changes in forest aboveground biomass stock and carbon fluxes, the direct effect of

climate on the seasonal carbon cycle of tropical forests remain unclear. Contrasting results have been reported depending on methods used. Studies show an increase of aboveground biomass gain in the wet season from direct measurement (biological field measurements), or, from indirect measurement, an increase of canopy photosynthetic capacity in the dry season (remote sensing, flux tower network) (Wagner et al., 2013). Several hypotheses have been proposed to explain these discrepancies: (i) wood productivity, estimated from trunk diameter increment, is mainly controlled by water availability (Wagner et al., 2014),

but seasonal variation in carbon allocation to the different parts of the plant (crown, roots) also contribute to optimizing resource use (Doughty et al., 2014, 2015); (ii) litterfall peak mainly occurs during dry periods as a combination of two potential climate drivers: seasonal changes in daily insolation leading to production of new leaves and synchronous abscission of old leaves, and high evaporative demand and low water availability that both induce leaf shedding in the dry season (Borchert et al., 2015; Zhang et al., 2014; Wright and Cornejo, 1990; Chave et al., 2010; Myneni et al., 2007; Jones et al., 2014; Bi et al., 2015); and

(iii) photosynthesis on a global scale is mainly controlled by water limitations and is sustained during the dry season above a threshold of 2000 mm of mean annual precipitation (Restrepo-Coupe et al., 2013; Guan et al., 2015).

Here, we determine the dependence of seasonal aboveground wood productivity, litterfall and canopy photosynthetic capacity (using the MODIS Enhanced Vegetation Index – EVI as a proxy) on climate across the tropics, and assess their interconnections in the seasonal carbon cycle.We use a unique satellite and ground-based combination of monthly data sets from

89 pan-tropical experimental sites (68 include aboveground wood productivity and 35 litter productivity measurements), their associate canopy photosynthetic capacity and climate to address the following questions: (i) Are seasonal aboveground wood productivity, litterfall productivity and photosynthetic capacity dependent on climate? (ii) Does a coherent pan-tropical rhythm exist among these three key components of forest carbon fluxes? (iii) if so, is this rhythm primarily controlled by exogenous (climate) or endogenous (ecosystem) processes?

We found that aboveground wood productivity and litterfall are directly related to climate seasonality and particularly to variations in precipitation and evaporation demand. Patterns of photosynthetic capacity are more complex as they respond positively to precipitation when mean annual precipitation is $< 2000$ mm.yr$^{-1}$ (water-limited sites) and to radiation otherwise (light-limited sites). Consequently, photosynthetic capacity and aboveground wood productivity have similar seasonal patterns in water-limited sites. In contrast, in light-limited forests, we observed decoupled seasonal patterns between aboveground wood




productivity and photosynthetic capacity, likely indicating an asynchrony in the use of photosynthesis products for aboveground wood productivity. Precipitation exerts a first-order control on the seasonality of canopy photosynthetic capacity and wood productivity. With reduction in mean annual precipitation, we found that the drivers of seasonality in canopy photosynthetic activity shifted from radiation to precipitation. Because of water scarcity in the dry season, water-limited forests are unable to

maintain maximum canopy photosynthetic throughout times of high solar radiation. This likely indicates an overall decrease in tropical forest productivity in a drier climate.

## 2   Methods

### 2.1   Datasets

We compiled the literature of publications reporting seasonal wood productivity of tropical forests. Seasonal tree growth

measurements in 68 pantropical forest sites, 14481 individuals, were obtained from published sources when available or directly from the authors (Table 2, Figures 1). The data set consists of repeated seasonal measurements of tree diameter mostly with dendrometer bands (94.1%), electronic point surveys (4.4%) or graduated tapes (1.5%). The names of all recorded species were checked using the Taxonomic Name Resolution Service and corrected as necessary (Boyle et al., 2013; Chamberlain and Szocs, 2013). Botanical identifications were made at the species-level for 11967 trees, at the genus-level for 1613 trees, family-level

for 171 trees and unidentified for 730 trees. Wood density values were taken from the Global Wood Density Database (Chave et al., 2009; Zanne et al., 2009) or from the authors when measured on the sample (Table 2). Direct determination for 455 trees and species mean was assumed for an additional 8671 trees. For the remaining 5355 trees, we assumed genus mean (4639), family mean (136) or site mean (580) of wood density values as computed from the global database (Zanne et al., 2009). Palms, lianas and species from mangrove environments were excluded from the analysis. Diameter changes were converted

to biomass estimates using a tropical forest biomass allometric equation – which uses tree height (estimated in the allometric equation if not available), tree diameter and wood density (Chave et al., 2014) – and then the mean monthly increment of the sample was computed for each sample. For each tree, unusual increments were identified and corrected when it was possible by replacing them with the mean increment of t+1 and t-1, or deleted. To detect the errors of overestimated or underestimated growth, increment histogram of each sites was plotted. For each suspect error, increment trajectory of trees were then visually

assessed to confirm the error. If the increment was identified as an error, it was corrected with linear approximation.

Seasonal litterfall productivity measurements from a previously published meta-analysis were used for South America (Chave et al., 2010) (description in Table 1 of (Chave et al., 2010)). In this dataset, we used only data with monthly measurements from old-growth forests, as some sites have plots of both secondary and old-growth forests; flooded forests were excluded. Additionally to these 23 sites, we compiled the seasonal leaf/litterfall data of 12 sites where we already had tree

growth measurements (Fig. 1 and Table 3). For these 35 sites, 26 had monthly leaf-fall and 9 had monthly litterfall data (leaf-fall, twigs usually less than 2 cm in diameter, flowers and fruits). The Pearson correlation coefficient between leaf-fall and litterfall for the 20 sites where both data are available is 0.945 (Pearson test, t = 42.7597, df = 218, p-value < 0.001). Consequently, we assumed that the seasonal pattern of litterfall is not different from seasonal pattern of leaf-fall.



Enhanced Vegetation Index (EVI) was used as a proxy for canopy photosynthetic capacity in tropical forest regions (Huete et al., 2006; Guan et al., 2015). EVI for the 89 experimental sites (Fig. 1) was obtained from the Moderate Resolution Imaging Spectroradiometer (MODIS) MCD43 product collection 5 (4 May 2002 to 30 September 2014). Before computing the mean monthly EVI per site, we did a pixel selection in five steps: (i) selection of all the pixels in a square of side 40 km, centered

on the pixel containing each site (6561 pixels per site); (ii) in this area, the pixels containing the same or at least 90% of the site land cover pixel were selected, based on MCD12Q1 for 2001–2012 at 500 m resolution (Justice et al., 1998); (iii) thereafter, only the pixels forested in 2000 and without loss of forest and with tree cover above or equal to the site tree cover were retained using using Global forest cover loss 2000–2012 and Data mask based on Landsat data (Hansen et al., 2013); (iv) only pixels with a range of $\pm$ 200 m the site altitude were retained, using NASA Shuttle Radar Topographic Mission

(SRTM) data, reprocessed to fill in the original no-data holes (Jarvis et al., 2008); (v) for corrected reflectance computation we used quality index from 0 (Good quality) to 3 (All magnitude inversions or 50% or less fill-values) extracted from MCD43A2. When required, data sets used to make the selection were aggregated to the spatial resolution of MCD43 product (500 m) and reprojected in the MODIS sinusoidal projection. The reflectance factors of red (0.620 - 0.670 $\mu$m, MODIS band 1), NIR (0.841 - 0.876 $\mu$m, MODIS band 2) and blue bands (0.459 - 0.479 $\mu$m, MODIS band 3) of the retained pixels were modeled with

the RossThick-LiSparse-Reciprocal model parameters contained in the MCD43A1 product with view angle $\theta_v$ fixed at 0°, sun zenith angle $\theta_s$ at 30° and relative azimuth angle $\Phi$ at 0° and EVI was computed as shown in Equation 1:

$$EVI \quad = \quad 2.5 \times \frac{NIR - red}{NIR + 6 \times red - 7.5 \times blue + 1} \tag{1}$$

To filter the time series, EVI above or below the 95% confidence interval of the site's EVI values were excluded. Then, the 16-days time series were interpolated to a monthly time step. Finally, the interannual monthly mean of EVI for each site was

20 computed. Further, the $\Delta\text{EVI}_{wet-dry}$ index was computed for each site, that is, the differences of wet- and dry-season EVI normalized by the mean EVI, where dry season is defined as months with potential evapotranspiration above precipitation (Guan et al., 2015). For the sites where evapotranspiration is never above precipitation, dry season was defined as months with normalized potential evapotranspiration above normalized precipitation. In this study $\Delta EVI_{wet-dry}$ computed from MODIS MCD43A1 is correlated with MOD13C1 (Amazonian sites: $\rho_{Spearman}$=0.90; pan-tropical sites: $\rho_{Spearman}$=0.86) and MAIAC

(Amazonian sites: $\rho_{Spearman}$=0.89) products (Supplementary Fig. S4).

To extract the monthly climate time series for the 89 experimental sites (Fig. 1), we used climate datasets from three sources: the Climate Research Unit (CRU) at the University of East Anglia (Mitchell and Jones, 2005), the Consortium for Spatial Information website (CGIAR-CSI, http://www.cgiar-csi.org) and from NASA (Loeb et al., 2009). From the CRU, we used variables from the CRU-TS3.21 monthly climate global dataset available at 0.5° resolution from 1901–2012: cloud cover

($cld$, unit: %); precipitation ($pre$, mm); daily mean, minimal and maximal temperatures (respectively $tmp$, $tmn$ and $tmx$, ° C); temperature amplitude ($dtr$, ° C); vapour pressure ($vap$, hPa); and potential evapotranspiration ($pet$, mm). The maximum climatological water deficit ($CWD$) is computed with CRU data by summing the difference between monthly precipitation and monthly evapotranspiration only when this difference is negative (water deficit) (Chave et al., 2014). From the CGIAR-CSI, we used the Global Soil-Water Balance, soil water content ($swc$, %) (Zomer et al., 2008). Additionally, we used monthly incoming



radiation at the top of the atmosphere ($rad$, W.m$^{-2}$) covering the period from 2000 to 2012 at 0.5° spatial resolution from the CERES instruments on the NASA Terra and Aqua satellites (Loeb et al., 2009). Additional to the temporal series of climate variables, we extracted the Global Ecological Zones ($GEZ$) of the sites. These GEZ are defined by the Food and Agriculture Organization of the United Nations (FAO) and relies on a combination of climate and (potential) vegetation (FAO, 2012).

5    To analyze only seasonality, the site effect was removed in all the datasets, that is, the monthly values were normalized by their site's annual mean values and standard deviation. The 89 sites represent a large sample of tropical forests under different tropical and subtropical climates corresponding to six global ecological tropical zones (FAO, 2012): Tropical rain forest (TAr, 41 sites), Tropical moist deciduous forest (TAwa, 23 sites), Tropical dry forest (TAwb, 14 sites), Tropical mountain systems (TM, 7 sites), Tropical shrubland (TBSh, 1 site) and Subtropical humid forest (SCf, 3 sites).

## 2.2    Data analysis

### 2.3    Effect of stem hydration on wood productivity

Changes in tree circumference with dendrometers are commonly used to characterize seasonal wood productivity. However, accelerated changes in circumference increments during the onset of the wet season can be caused by bark swelling as they become hydrated (Stahl et al., 2010). Similarly, bark shrinking during dry periods can mask any secondary growth and even lead

to negative growth increments (Stahl et al., 2010; Baker et al., 2002). Stem shrinkage during dry periods may be an important limitation of this work (Sheil, 2003; Stahl et al., 2010), as negative monthly growth values exist at almost all the study sites. Since the measurements are stem radius or circumference changes rather than wood formation, it is difficult to distinguish between true wood formation and hydrological swelling and shrinking. Direct measurements of cambial growth like pinning and microcoring currently represent the most reliable techniques for monitoring seasonal wood formation; however, all these meth-

ods are highly time-consuming, which severely restricts their applicability for collecting large data sets (Makinen et al., 2008; Trouet et al., 2012). Nevertheless, some observations already exist to compare growth from dendrometers and cambial growth at a seasonal scale for the same trees. In a tropical forest in Ethiopia experiencing a strong seasonality, high-resolution electronic dendrometers have been combined with wood anatomy investigation to describe cambial growth dynamics (Krepkowski et al., 2011). These authors concluded that water scarcity during the long dry season induced cambial dormancy (Krepkowski

et al., 2011). Furthermore, after the onset of the rainy season, (i) bark swelling started synchronously among trees, (ii) bark swelling was maximum after few rainy days, and (iii) evergreen trees were able to quickly initiate wood formation. In a laboratory experiment of trunk section desiccation, Stahl et al. (2010) have showed a decrease in the diameter of the trunk sections ranging from 0.08% to 1.73% of the initial diameter and significantly correlated with the difference in water content in the bark, but not with the difference in water content in sapwood. The variation in the diameter of the trunk sections were observed

when manipulating the chamber relative air humidity from 90% to 40%. However, these values are not representative of the *in situ* French Guiana climatic conditions, which is where the trunk sections have been collected and where relative humidity never falls below 70%. Negative increments were reported for one-quarter of their sample with dendrometers measurements in the field. Recently, at the same site, some authors showed that biomass increments were highly correlated between the first and



last quantiles of trunk bark thickness and between the first and the last quantile of trunk bark density, thereby suggesting that secondary growth is driven by cambial activity (Wagner et al., 2013) and not by water content in bark. At Paracou, a recent study showed a decrease or stop in the cambial growth for some species during the dry season, based on analysis of tree rings (Morel et al., 2015).

In a temperate forest, Makinen et al. (2008) simultaneously using dendrometer pinning and microcoring on Norway spruce and Scots pine, (see Fig. 3 and Fig. 5 in (Makinen et al., 2008)) showed that a lag of two weeks exists between the growth measured by dendrometers, but the general pattern of growth is highly correlated. Furthermore, a substantial rainfall event occurring after the end of the cambial growth season did not induce xylem initiation or false ring formation Trouet et al. (2012); Wagner et al. (2012). In La Selva (Costa Rica) where there is no month with precipitation below 100 mm, a seasonal

variation is reported, thereby suggesting a seasonality only driven by cambial growth. In conclusion, swelling and shrinking exist and could result from different biotic and abiotic causes, cell size, diameter, bark thickness and relative air humidity (Stahl et al., 2010; Baker et al., 2002). To test how swelling and shrinking affect our results, we made first the analysis with all the data, and then a second analysis discarding the first month of the wet season (first month with precipitation $> 100$ mm) and the first month of the dry season (precipitation $< 100$ mm). Here, we assume that swelling occurs in the first month of the wet

season and shrinking occurs in the first month of the dry season, as already observed. Removing the first month of dry season and wet season (defined respectively as the first month with precipitation $> 100$ mm and the first month with precipitation $< 100$ mm) did not affect the results of the predictive model of wood productivity by precipitation, that is, intercepts and slopes are not significantly different in both models (overlaps of the 95% confidence interval of coefficients and parameters, Table 4).

## 2.4  Seasonality analysis

To address the first question 'Are seasonal aboveground wood productivity, litterfall productivity and photosynthetic capacity dependent on climate?', we analyzed with linear models the relationship between our variable of interest and each climate variable at each site and at t, t-1 month and t+1 month. These lags were chosen to account artificially for variations in the climate seasonality. The results were classified for each variable as a count of sites with significantly positive, negative or not significant results. To enable comparison, if the overall effect of the climate variable was negative, the linear model for each

site was run with the climate variable multiplied by -1. For a given climate variable, a site with a significant association at only one of the time lag (-1, 0 or 1) was classified as significant. Then, a McNemar test was run to compare the proportion of our classification (negative, positive or no relationship) between all paired combinations of climate variables accounting for dependence in the data, that is, to compare not only the proportion of positive, negative and no significant effect between two climate variables but also to detect if the sites in each of the classes were similar. To determine which climate variables

explain the same part of variance and to enable interpretation, a cluster analysis was performed on the table of p-values of the McNemar test using ward distance.

When the climate variable with direct effect was identified, we built a linear model to predict wood and litter productivity seasonality with climate in all sites. For EVI, two climate variables were identified and their influence was dependent on the site values of $\Delta \text{EVI}_{wet-dry}$. To find the $\Delta \text{EVI}_{wet-dry}$ threshold of main influence of each variable, the $R^2$ of the linear relationship





EVI as a function of the climate variable for different values of $\Delta$EVI$_{wet-dry}$ threshold were computed. R$^2$ was computed for the sample above or below $\Delta$EVI$_{wet-dry}$ depending on the relationship of each variable to the threshold. The optimal threshold of $\Delta$EVI$_{wet-dry}$ for climate variable influence on normalized EVI was defined by a break in the decrease of R$^2$ values. Optimal thresholds were then used to define the range of $\Delta$EVI$_{wet-dry}$ where EVI is influenced by one of the climate

variables, the other and by both. To find the best linear combination of variables that contains the maximum information to predict EVI, we ran an exhaustive screening of the candidate models with the identified climate variables and their interactions with the $\Delta$EVI$_{wet-dry}$ classes using a stepwise procedure based on the Bayesian information criterion, BIC (Schwarz, 1978).

To address the second question 'Does a coherent pan-tropical rhythm exist among these three key components of the forest carbon fluxes?', we analyzed the linear relationship between wood, litter productivity and canopy photosynthetic capacity. The

non-parametric Mann-Whitney test was used to determine the association between wood/litter productivity and photosynthesis rhythmicity depending on site limitations.

To address the third question 'Is the rhythm among these three key components of the forest carbon controlled by exogenous (climate) or endogenous (ecosystem) processes?', we analyzed the linear relationship between $\Delta$EVI$_{wet-dry}$ and mean annual precipitation, as well as the relationship between $\Delta$EVI$_{wet-dry}$, $\Delta$wood productivity$_{wet-dry}$ and $\Delta$litter productivity$_{wet-dry}$

and maximum climatological water deficit ($CWD$). $\Delta$EVI$_{wet-dry}$, $\Delta$wood productivity$_{wet-dry}$ and $\Delta$litter productivity$_{wet-dry}$ indices are the differences of wet- and dry-season variable values normalized by the mean of the variable, where the dry season is defined as months with potential evapotranspiration above precipitation.

To avoid over-representation of sites with the 'same climate' (that is, to account for spatial and temporal autocorrelation in the climate data) cross correlation (positive and negative) were computed within sites for the monthly climate variables $rad$,

$pre$, $pet$, $dtr$, $tmn$ and $tmx$. The site's annual values of the same climates variable were added in the table. After scaling and centering the table, the Euclidian distance between each site and the mean table of all other sites (baricenter) was computed. We defined the weight of each site as the distance to the other divided by the maximum distance to the other. This distance was used as a weight in the linear models.

All analysis were performed in R (Team, 2014).

# 3 Results

## 3.1 Climate footprint in seasonal carbon assimilation and storage

A direct and dominant signal of precipitation seasonality was found in seasonality of wood productivity for 59 out of the 68 sites (86.8%) where wood productivity data were available (cluster of variables in Fig. 2a with temperature amplitude ($dtr$), cloud cover ($cld$), precipitation ($pre$) and soil water content ($swc$), Methods 2.2 and Supplementary Table S1). All the variables

in this cluster are wet season indicators: low temperature amplitude, high precipitation, high soil water content and high cloud cover. Two other clusters of climate variables are apparently associated with wood productivity. However, the climate variables that better explained wood productivity in these two clusters, vapor pressure ($vap$) and mean temperature ($tmp$), respectively, are highly correlated with precipitation in the clusters (Fig. 2a and Supplementary Table S3-S4). In spite of this dominant





signal, these are outliers in our data, that exhibit no relationship or a negative relationship with precipitation (Appendix A1). Four of the five sites that have no dry season (months with precipitation below 100 mm) were amongst these outliers.

It is interesting to note that 48.0% of the monthly wood productivity is explained by the single variable 'precipitation' (model $m_{WP}$ in Table 1). The linear model with monthly precipitation only ($m_{WP}$) was able to reproduce the seasonality of

5 the majority of the sites analyzed (Fig. 3a). No monthly lag between predicted and observed seasonality was observed for 35 sites. For 63 sites, a lag between -2 and +2 months was observed (Fig. 4a).

Canopy photosynthetic capacity, as estimated by EVI, for the 89 experimental sites, displayed an intriguing pattern with monthly precipitation, apparently related to the difference of $\Delta EVI_{wet-dry}$ (Fig. 5a), an indicator of the dry season evergreen state maintenance (Guan et al., 2015), computed as the difference between the mean EVI of the wet season ($pre \geq pet$) and

10 of the dry season ($pre < pet$) (Methods 2.1). This pattern can be explained by a change in the climate parameters that mainly control photosynthesis, from precipitation in water-limited sites ($\Delta EVI_{wet-dry} > 0.0378$, Fig. 5b) to maximal temperature in light-limited site ($\Delta EVI_{wet-dry} < -0.0014$, Fig. 5c and Supplementary Fig. S1). Sites with mixed influence of precipitation and temperature are found between the range of $\Delta EVI_{wet-dry}$ [-0.0014;0.0378] (Fig. 6 for the definition of the thresholds). In our sample, the shift in climate control depends on the annual water availability. That is, sites are not water-limited above

15 2000 mm.yr$^{-1}$ of mean annual precipitation (Fig. 5d), as previously observed (Guan et al., 2015), but then they are light-limited as shown by the relationship between photosynthetic capacity and maximal temperature (Fig. 5c). Light-limited sites are located in Amazonia, in the south of Brazil and in Southeast Asia (Fig. 8). For these sites, while solar radiation at the top of the atmosphere is not different between the dry and wet seasons, maximal temperature is higher in the dry season, thereby reflecting solar energy available for the plants (Fig. 7). With the model mBIC$_{EVI}$ (Table 1), precipitation, maximal

temperatures and their thresholds explained 54.8% of the seasonality of photosynthetic capacity (Fig. 3c). For 39 sites, no seasonal lag between predicted and observed seasonality of canopy photosynthetic capacity was observed using the model mBIC$_{EVI}$. However, a majority of the sites (82 sites) appeared to have a lag between -2 and +2 months (Fig. 4c). The model failed to reproduce the seasonality for seven sites (one water-limited, one light-limited and five mixed sites).

For 27 out of the 35 sites (77.1%) where litter data were available, litter productivity was associated with dry season indica-

25 tors (lack of precipitation, high evaporation, low soil water content and high temperature amplitude, Fig. 2b). Surprisingly, we found that cloud cover ($cld$), an indirect variable, was the best single predictor of litterfall seasonality (Table 1). Direct effects are observed only for potential evapotranspiration ($pet$) and temperature amplitude ($dtr$) (Fig. 2b and Supplementary Table S5). A second cluster of climate variables is associated with litter productivity but a key variable in this subgroup, minimal temperature ($tmn$), is correlated with cloud cover (Supplementary Table S7). Despite this dominant signal, outliers showing

no relationship with $cld$ exist in our data (Appendix A2). The predictive model with cloud cover as a single variable (Table 1) explains 31.7% of the variability and performs well to reproduce the seasonality of litterfall productivity (Fig. 3b and 4b).

At a pan-tropical scale, 48% of the variability of monthly aboveground wood productivity (Fig. 3a and Table 1) and 31.7% of the monthly litterfall seasonality can be linearly explained with a single climate variable (Fig. 3b). The relationship between photosynthetic capacity (EVI) and climate is more complex; however, 54.8% of the monthly EVI variability can be linearly

explained with only two climate variables, precipitation and maximal temperature (Fig. 3c).




## 3.2 Decoupling wood productivity, litter productivity and canopy photosynthetic capacity seasonality

In sites where both measurements were available, we observed a negative relationship between wood productivity and litterfall (Fig. 9, supported by linear analysis, Supplementary Fig. S2). This relationship is consistent across the tropics and constant for all our sites (Fig. 10c), independently of the site water or light limitations (Mann-Whitney test, U = 746, p-value = 0.0839).

Wood productivity and litterfall are mainly driven by only one climate driver in our results, precipitation and cloud cover respectively. The seasonality of these climate drivers are coupled for all the sites, where maximum precipitation occurs in the wet season while minimum cloud cover occurs in the dry season.

EVI seasonality is well associated with aboveground wood production for water-limited forests, as a consequence of their relationship with precipitation (Fig. 10a). However, aboveground wood production is better explained by precipitation than EVI ($R^2$ of 0.503 and 0.451 respectively).

Conversely, in light-limited sites and forests with mixed limitations (mixed forests), EVI is weakly coupled with the seasonality of wood productivity (respectively p-value = 0.0633, $R^2$ = 0.017 and p-value = 0.0124, $R^2$ = 0.055). Therefore, we conclude that the relationship between EVI and wood productivity depends on site limitations (Mann-Whitney test, U = 874.5, p-value = 0.0012).

The relationship between EVI and litter production is not constant (Fig. 10b), and also depends on site limitations (Mann-Whitney test, U = 1016.5, p-value < 0.001). EVI is consistently negatively associated with litterfall production for water-limited forests (p < 0.001, $R^2$ = 0.510), reflecting forest 'brown-down' when litterfall is maximal. Litter production is slightly better explained by cloud cover than EVI ($R^2$ of 0.533 and 0.510 respectively) and they predict the same effect for the same site (McNemar test, p-value = 0.999). No significant associations are found between EVI and litter in forests with mixed limitations (p-value = 0.8531, $R^2$ < 0.0001) and in light-limited forests (p-value = 0.4309, $R^2$ < 0.0001).

$\Delta EVI_{wet-dry}$ and $\Delta$wood productivity$_{wet-dry}$ are dependent on annual water availability (Fig. 11a-b and Fig. 5d). $\Delta$wood productivity$_{wet-dry}$ is close to zero and could be negative for light-limited sites; the amplitude of the seasonality is driven by the annual water availability. The values for $\Delta$wood productivity$_{wet-dry}$ in South East Asia are all negative. This is consistent with the negative or null associations of wood productivity and precipitation at these sites (Appendix A1). $\Delta$litter productivity$_{wet-dry}$ is poorly correlated with maximum climatological water deficit ($CWD$).

## 4 Discussion

We have found a remarkably strong climate signal in the seasonal carbon cycle components studied across tropical forests. While wood and litterfall production appear to be dependent on a single major climate driver across the tropics (water availability), the control of photosynthetic capacity varies according to the increase in annual water availability, shifting from water-only to light-only drivers.

Minimum aboveground wood production tends to occur in the dry season. This result is not new (Wagner et al., 2013), but here we confirm this pattern. From the climatic point of view, months with the lowest water availability are less favourable for cell expansion, as water stress is known to inhibit this process, as observed in dry tropical sites (Borchert, 1999; Krepkowski





et al., 2011). This pattern is found in water-limited, mixed and light-limited sites. At the very end of the water availability gradient (wettest ones), some sites have no relationship or a negative relationship with monthly precipitation, as observed in Lambir, Malaysia (Kho et al., 2013). These sites, three in South East Asia and one in South Brazil, have no marked dry season, defined as months with precipitation below 100 mm. These relationships with monthly precipitation could reflect cambial

dormancy induced by soil water saturation, as observed in Amazonian floodplain forests (Schöngart et al., 2002), and/or be related to limited light availability due to persistent cloud cover. However, for these ultra wet sites, the lack of field data limits the analysis of the effects of climate on the seasonality of aboveground wood production.

Maximum litterfall, for most of our sites, occurs during the months of minimum cloud cover during the dry season. It is known that the gradient from deciduous to evergreen forests is related to water availability, with the evergreen state sustained

during the dry season above a mean annual precipitation threshold of approximately 2000 mm.yr$^{-1}$ (Guan et al., 2015). The litterfall peak occurs when evaporative demand is highest. The maintenance of litterfall seasonality in the light-limited sites could be driven mostly by a few large/tall canopy trees shedding leaves, mainly in response to high evaporative demand. This can explain why litterfall occurs in the dry season and is decoupled from EVI, a parameter that integrates the entire canopy (Fig. 10b). On the other hand, in water-limited sites, most of the trees shed their leaves, thereby resulting in a litterfall signal

coupled with EVI 'brown-down' (Fig. 10b).

Canopy photosynthetic capacity has different climate controls depending on water limitations (Fig. 5). As already observed, in sites with mean annual precipitation below 2000 mm.yr$^{-1}$ (Fig. 5d), photosynthetic capacity is highly associated with water availability (Guan et al., 2015) and highly dependent on monthly precipitation (Fig. 5b). This seems to confirm that longer or more intense dry seasons can lead to a dry-season reduction in photosynthetic rates (Guan et al., 2015). In addition to

the control by water availability (Guan et al., 2015; Bowman and Prior, 2005; Hilker et al., 2014), we demonstrated that for sites where water is not limiting, photosynthetic capacity depends on maximal temperatures, which reflects available solar energy or daily insolation at the forest floor (Fig. 7). For these sites, the EVI peak occurs at the same time as the maximal temperature peak, which supports the hypothesis of the detection of a leaf flushing signal induced by a preceding increase of daily insolation (Borchert et al., 2015). This result is also consistent with flux-tower-based GPP estimates in neotropical forests

(Restrepo-Coupe et al., 2013; Guan et al., 2015; Bonal et al., 2008). If the increase in EVI is a proxy of leaf production, our result supports the satellite-based hypothesis that temporal adjustment of net leaf flush occurs to maximize water and radiation use while reducing drought susceptibility (Myneni et al., 2007; Jones et al., 2014; Bi et al., 2015).

We demonstrated that the seasonality of aboveground wood production and litterfall are coupled while photosynthetic capacity seasonality can be decoupled from wood and litterfall production seasonality depending on the local water availability

(Fig. 10).

Further, our results show that carbon allocation to wood is prioritized in the wet season, independently of the site conditions (water- or light-limited). This priority has also been shown in forests impacted by droughts, where trees prioritized wood production by reducing autotrophic respiration even when photosynthesis was reduced as a consequence of water shortage (Doughty et al., 2015). However, there is still a lack of information on a wider scale regarding how trees prioritize the use

of non-structural carbohydrates. The potential decoupling of carbon assimilation and carbon allocation found here seems




to indicate a complex and indirect mechanism driving carbon fluxes in the trees. Some experimental results showed that endogenous and phenological rhythms can define the prioritization in carbon allocation and may be more important drivers of the carbon cycle seasonality than climate in tropical forests (Malhi et al., 2014; Doughty et al., 2014; Morel et al., 2015). This corroborates other results that indicate that growth is not limited by carbon supply in tropical forests (Körner, 2003; van der Sleen et al., 2015; Wurth et al., 2005). However, even if these results are in accordance with our results for light-limited sites, it must be noted that they cannot be generalized to water-limited sites, where climate constrains both photosynthetic capacity and wood productivity.

Canopy photosynthetic capacity and aboveground wood production appear to be predominantly driven by climate at seasonal and annual scales, thereby suggesting exogeneous drivers (Fig. 5 and Fig. 11). However, if litterfall was driven by climate only, its pattern would be more predictable, with a linear relationship between annual water availability ($CWD$) and $\Delta$litter productivity$_{wet-dry}$ such as for wood production (Fig. 11b-c), which would translate into a massive peak in the dry season. Even with the litterfall peak occurring mainly in the dry season, another part of the variation seems to be related to endogeneous drivers. Such endogeneous effects have already been observed in tropical forests, for example, seasonality of root production prioritized over leaf production in a dry site in Bolivia or leaf production occurrence during wet months in French Guiana (Doughty et al., 2014; Morel et al., 2015). If the molecular mechanisms of photoperiodic control of tree development are the same in temperate and tropical trees (Borchert et al., 2015), tropical tree phenology could depend on the following genetic loci: FLOWERING LOCUS T1 (FT1), FLOWERING LOCUS T2 (FT2) and EARLY BUD-BREAK 1 (EBB1), respectively for reproductive onset, vegetative growth and inhibition of bud set, and release from seasonal dormancy and bud break initiation (Yordanov et al., 2014; Hsu et al., 2011; Srinivasan et al., 2012). The lag between peak of litterfall in dry season and minimum photosynthetic capacity of the canopy we observe for light-limited sites (Fig. 10b) could reflect a mixture of bud sets and bud breaks with a relative weak synchronism due to the high diversity of species involved and the weakness of the seasonal signal of solar insolation. Our results are consistent with a seasonal cycle timed to the seasonality of solar insolation, but with an additional noise due to leaf renewal and/or net leaf abscission during the entire year unrelated to climate variations (Borchert et al., 2015; Myneni et al., 2007; Jones et al., 2014; Bi et al., 2015). While photosynthetic capacity and wood productivity appear mostly exogenously driven, litterfall is the result of both exogenous and endogenous processes.

In this study, we use EVI as an index of seasonality of canopy photosynthetic capacity based on the previously demonstrated correlation between canopy photosynthetic capacity from the MODIS sensor and solar-induced chlorophyll fluorescence (SIF) at a pan-tropical scale (Guan et al., 2015) and from the correlation between $\Delta EVI_{wet-dry}$ from MODIS MOD13C1, MCD43A1 and MAIAC products (Supplementary Fig. S4). Here, we show how satellite and field data can be used to infer characteristics of tropical forests carbon cycle in a consistent framework. To go further, it is necessary to determine the real amount of photosynthetic products in order to describe quantitatively the seasonal carbon cycle in tropical forests.





## 5 Conclusions

In summary, the seasonality of carbon assimilation and allocation through photosynthetic capacity and aboveground wood production is consistently and directly related to climate in tropical forested regions. Notably, we found that regions without annual water limitations exhibit a decoupled carbon assimilation and storage cycle, which highlight the complexity of carbon

5 allocation seasonality in the tropical trees. Although carbon assimilation is driven by water, whether the photosynthetic capacity seasonal pattern is driven by light or water depends on the limitations of site water availability. The first-order precipitation control likely indicates a decline in tropical forest productivity in a drier climate, by a direct limitation of canopy photosynthetic capacity in water-limited forests and, in light-limited forests, by a reduction of canopy photosynthetic capacity in the dry season.

### Appendix A: Description of outliers

### 10 A1 Wood productivity outliers

Although this dominant signal, outliers exist in our data showing negative (3 sites) or no relationship (6 sites) with precipitation. Due to the correlation of climate variables at the site scale, it is difficult to interpret each site alone; however, some groups arose in these outlier sites. The first group, the two sites Itatinga and Pinkwae, contains only saplings measurements. The second group, the sites with no month with precipitation below 100 mm, includes Lambir (Malaysia), Muara Bungo (Indonesia),

Pasoh (Malaysia), Flona SFP (Brazil). The third group includes two mountain sites, Tulua and Munessa. For Munessa, there is evidence of cambial growth related to precipitation Krepkowski et al. (2011); however, the sample we used comprises two species known to have different sensitivity to rainfall. The monthly mean of the sites' wood productivity could be responsible for the lack of rainfall-related pattern. Finally, for Caracarai (Brazil), there was a lack of six-month data encompassing the beginning and middle of the wet season, which has been linearly interpolated to the month; however, due to the important

sampling effort, we initially chose to keep this dataset.

### A2 Litterfall productivity outliers

Only one site, BDFFP, showed no apparent relationship between litter productivity and cloud cover (Supplementary Fig. S3). This site is in a fragmented forest where fragmentation is known to affect litterfall (Vasconcelos and Luizão, 2004). For the other outlier, they all have a peak of litterfall correlated with $pet$ or $cld$ (Supplementary Fig. S3). Three different groups can

be observed: (i) sites which have another peak of litterfall during the year (Cueiras, La Selva, Gran Sabana), (ii) sites with very skew litterfall peaks followed by an important decrease in litterfall, while the climate conditions are optimal for litterfall productivity from the viewpoint of the linear model (Capitao Paco, Rio Juruena and RBSF) and (iii) sites which have two peaks of $pet$, but litterfall occurs only during one of them (Apiau Roraima, Gran Sabana).





*Author contributions.* F.H.W., L.E.O.C.A., B.H., D.B. and C.S. wrote the paper, F.H.W., L.E.O.C.A. and B.H. conceived and designed the study, F.H.W. assembled the data sets, B.B. and J.V. contributed to the programing part, F.H.W. carried out the data analysis. All co-authors collected field data and commented on or approved the manuscript.

*Acknowledgements.* This project and F.H.W. have been funded by the Fapesp (Fundação de Amparo à Pesquisa do Estado de São Paulo,
processo 13/14520-6). J.P.L. and M.M.T. were funded by the CNPq and the FAPEMIG. B.P.M. was funded by the Australian Research Council for the project "Understanding the impact of global environmental change on Australian forests and woodlands using rainforest boundaries and Callitris growth as bio-indicators", grant number: DP0878177. A.B. was funded by the German Research Foundation (DFG) for the project BR1895/15 and the projects BR1895/14 and BR1895/23 (PAK 823). F.A.C. and J.M.F. were funded by the CNPq (grant 476477/2006-9) and the Fundação O Boticário de Proteção a Natureza (grant 0705-2006). F.R.C.C. was funded by the CNPq/PELD
"Impactos antrópicos no ecossistema de floresta tropical - site Manaus", Processo 403764/2012-2. J.G. was supported from the US Forest Service-International Institute of Tropical Forestry. A.D.G. funding was provided through ARC Linkage (Timber harvest management for the Aboriginal arts industry: socio-economic, cultural and ecological determinants of sustainability in a remote community context, LP0219425). S.F.O. was funded by the National Science Foundation BE/CBC: Complex interactions among water, nutrients and carbon stocks and fluxes across a natural fertility gradient in tropical rain forest (EAR 421178) and National Science Foundation Causes and implications of dry
season control of tropical wet forest tree growth at very high water levels: direct vs. indirect limitations (DEB 842235). E.E.M. was funded by the Academy of Finland (project: 266393). L.M. was funded by a grant provided by the European Union (FP6, INCO/SSA) for a two year (2006-2008) Project on management of indigenous tree species for restoration and wood production in semi-arid miombo woodlands in East Africa (MITMIOMBO). F.V. was supported by the German Research Foundation (DFG) by funding the projects BR 1895/14-1/2 (FOR 816) and BR 1895/23-1/2 (PAK 823). L.K.K. was supported by the Malaysian Palm Oil Board. D.M.D. was funded by the Hermon
Slade Foundation (Grant HSF 09/5). Data recorded at Paracou, French Guiana, were partly funded by an "Investissement d'Avenir" grant from the ANR (CEBA: ANR-10-LABX-0025). H.A.M. and J.J.C. thank the staff of the Jardín Botánico 'Juan María Céspedes' (INCIVA, Colombia) and the Instituto Boliviano de Investigación Forestal (IBIF, Bolivia) for their support, particularly to M. Toledo and W. Devia; and P. Roosenboom (INPA Co.) and his staff at Concepción (G. Urbano) for their help in Bolivia. H.A.M. and J.J.C. were funded by the following research projects "Análisis retrospectivos mediante dendrocronología para profundizar en la ecología y mejorar la gestión de los
bosques tropicales secos" (financed by Fundación BBVA) and "Regeneración, crecimiento y modelos dinámicos de bosques tropicales secos: herramientas para su conservación y para el uso sostenible de especies maderables" (AECID 11-CAP2-1730, Spanish Ministry of Foreign Affairs). C.S.L. was funded by a grant from FAPESP (Proc. 02/ 14166-3), and Brazilian Council for Superior Education, CAPES. J.H. was funded by two grants from the Deutsche Forschungsgemeinschaft (DFG): BR379/16 and HO3296/4. D.A.C. was funded by the U.S. National Science Foundation (most recently EAR0421178 & DEB-1357112), the U.S. Department of Energy, the Andrew W. Mellon Foundation, and
Conservation International's TEAM Initiative. C.S. was funded by a grant from the "European Research 991 Council Synergy", grant ERC-2013-SyG-610028 IMBALANCE-P. M.R.K., J.E.F.M., T.L.S and F.G. were funded by Petrobras SA. We further thank Jeanine Maria Felfili and Raimundo dos Santos Saraiva who contributed to this work but who are no longer with us.





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




**Table 1.** Intercepts and slopes of the fitted linear models for seasonal wood production ($m_{WP}$), litterfall ($m_{lit}$) and EVI ($mBIC_{EVI}$); with the seasonal climate variables: precipitation ($pre$), cloud cover ($cld$) and maximal temperature ($tmx$). Light-, water- and mixed limitation indicate the limitation of the sites and are defined with the value of $\Delta EVI_{wet-dry}$ (Fig. 6 for the definition of the thresholds).

| Model | Components | Coefficient (std. error) | t value | p-value | $R^2$ |
|---|---|---|---|---|---|
| Wood production ($m_{WP}$) | Intercept | 0.0005 (0.0249) | 0.02 | 0.9833 | 0.480 |
| | Precipitation | 0.6869 (0.0260) | 26.40 | <0.0001 | |
| Litterfall ($m_{lit}$) | Intercept | 0.0000 (0.0389) | 0.00 | 0.9999 | 0.317 |
| | Cloud cover | -0.5685 (0.0407) | -13.98 | <0.0001 | |
| EVI ($mBIC_{EVI}$) | Intercept | 0.0000 (0.0197) | 0.00 | 0.9999 | 0.548 |
| | Maximal temperature in light-limited sites | 0.7643 (0.0396) | 19.28 | <0.0001 | |
| | Maximal temperature in sites with mixed limitations | 0.1683 (0.0545) | 3.09 | 0.0020 | |
| | Maximal temperature in water-limited sites | -0.1100 (0.0275) | -4.00 | <0.0001 | |
| | Precipitation in sites with mixed limitation | 0.3697 (0.0545) | 6.78 | <0.0001 | |
| | Precipitation in water-limited sites | 0.8149 (0.0275) | 29.60 | <0.0001 | |

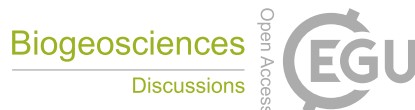



**Table 2.** Description of the study sites. For each site, continent (Africa – Af, America – Am, Asia – As and Australia – Aus), country, full site name and geographical coordinates (long.-lat., in degrees) are reported. The next column reports annual litterfall measurement of wood productivity and litterfall (WP+LT) or only wood productivity (WP), the time scale of the measurements, the number of species, the number of trees, the number of species, the reference for the wood densities, the period of the measurements, the mean diameter (mm) of the sample and the mean wood productivity in kg.tree$^{-1}$.year$^{-1}$.

| reference | cont | country | site | Lat | Lon | type | method | time_scale | N_tree | N_sp | wsg | duration | diam | dagb±SE |
|---|---|---|---|---|---|---|---|---|---|---|---|---|---|---|
| Detienne and A. (1976) | Af | Cameroon | MBalmayo | 3.515 | 11.501 | WP | DB | bi-weekly | 1 | 1 | Zanne et al. (2009) | 1/1966-12/1970 | 491.8 (491.8-491.8) | 41.24±4.698 |
| Detienne and A. (1976) | Af | CAR | MBaiki | 3.812 | 17.881 | WP | DB | bi-weekly | 1 | 1 | Zanne et al. (2009) | 2/1969-11/1970 | 282.9 (282.9-282.9) | 9.51±1.651 |
| Detienne and A. (1976) | Af | CAR | Mokindu | 3.650 | 18.350 | WP | DB | bi-weekly | 1 | 1 | Zanne et al. (2009) | 2/1969-12/1970 | 391.1 (391.1-391.1) | 11.52±2.771 |
| Couralet et al. (2010) | Af | DRC | Luki forest | -5.583 | 13.183 | WP | DB | monthly | 40 | 4 | Zanne et al. (2009) | 4/2006-8/2007 | 243.2 (121.4-456.9) | 12.23±1.646 |
| Krepkowski et al. (2011) | Af | Ethiopia | Munessa | 7.433 | 38.867 | WP | EPD | 30-min | 9 | 2 | Zanne et al. (2009); Aerts (2008) | 3/2008-12/2012 | 327 (168.3-582.1) | 11.5±1.309 |
| Baker et al. (2003) | Af | Ghana | Bonsa River | 5.333 | -1.850 | WP | DB | monthly | 36 | 2 | Zanne et al. (2009) | 8/1997-12/1999 | 380.7 (107.2-824.3) | 20.18±0.976 |
| Swaine et al. (1990) | Af | Ghana | GPR | 5.908 | 0.061 | WP | DB | monthly | 12 | 7 | Zanne et al. (2009) | 1/1978-4/1979 | 112.4 (45.7-186.6) | 1.05±0.655 |
| Lieberman (1982) | Af | Ghana | Pinkwae | 5.750 | -0.133 | WP | DB | monthly | 7 | 2 | Zanne et al. (2009) | 3/1978-4/1979 | 51.7 (34.8-91.7) | 0.21±0.188 |
| Baker et al. (2003); Owusu-Sekyere et al. (2006) | Af | Ghana | Tinte Bepo | 7.067 | -2.100 | WP+LP | DB | monthly | 40 | 3 | Zanne et al. (2009) | 7/1997-1/1999 | 346.6 (172.9-780.5) | 20.71±1.498 |
| Devineau (1991) | Af | Ivory Coast | Lamto | 6.217 | -5.033 | WP | DB | monthly | 23 | 13 | Zanne et al. (2009) | 7/1972-12/1981 | 168.6 (74.3-322.5) | 3.74±0.231 |
| Detienne and A. (1976) | Af | Ivory Coast | Oume | 6.383 | -5.416 | WP | DB | bi-weekly | 1 | 1 | Zanne et al. (2009) | 4/1966-12/1970 | 550.4 (550.4-550.4) | 25.12±3.806 |
| Glinias et al. (2013) | Af | Kenya | Kakamega | 0.258 | 34.883 | WP | DB | monthly | 766 | 52 | Zanne et al. (2009); Becker et al. (2012) | 6/2003-12/2009 | 355 (98.3-1624.7) | 11.99±0.108 |
| Elifuraha et al. (2008) | Af | Tanzania | Kitulangalo | -6.667 | 37.973 | WP | DB | monthly | 53 | 10 | Zanne et al. (2009) | 2/2007-8/2008 | 237.1 (71-632.3) | 4.27±1.239 |
| Glinias et al. (2013) | Af | Uganda | Budongo | 1.750 | 31.500 | WP | DB | monthly | 312 | 64 | Zanne et al. (2009); Becker et al. (2012) | 1/2005-12/2009 | 230.7 (93.7-1163.8) | 4.22±0.115 |
| Chidumayo (2005) | Af | Zambia | Makeni | -15.467 | 28.183 | WP | DB | monthly | 45 | 4 | Zanne et al. (2009) | 12/1996-6/2003 | 69.7 (28.2-167.7) | 13.68±0.633 |
| Chidumayo (2005) | Af | Zambia | UNZA | -15.392 | 28.333 | WP | DB | monthly | 51 | 2 | Zanne et al. (2009) | 1/1997-5/2002 | 68.6 (30.7-340) | 6.88±0.329 |
| Mendivelso et al. (2013) | Am | Bolivia | Inpa | -16.117 | -61.717 | WP | DB | monthly | 43 | 5 | Mendivelso et al. (2013) | 8/2010-9/2011 | 162.5 (107.7-290.7) | 3.67±0.58 |
| Dünisch et al. (2002) | Am | Brazil | Aripuana | -10.150 | -59.433 | WP | DB | monthly | 60 | 2 | Zanne et al. (2009) | 10/1998-10/2001 | 413.3 (138.3-1120.4) | 45.43±1.442 |
| Chagas et al. (2004) | Am | Brazil | Caetetus | -22.400 | -49.700 | WP | DB | monthly | 70 | 7 | Zanne et al. (2009) | 2/1996-7/1997 | 203.2 (50.9-651) | 5.91±0.89 |
| Castilho et al. (2012) | Am | Brazil | Caracarai | 1.476 | -61.019 | WP+LP | DB | 3-monthly | 2396 | 202 | Zanne et al. (2009); Bouanerges (2012) | 1/2013-3/2014 | 198.6 (34.3-1049.6) | 4.55±0.105 |
| Melgaço (2014) | Am | Brazil | Ducke | -2.952 | -59.944 | WP+LP | DB | bi-monthly | 1972 | 540 | Zanne et al. (2009) | 2/2013-2/2014 | 266.1 (97.3-1367.9) | 11.67±0.266 |
| Lisi et al. (2008); Ferreira-Fedele et al. (2004) | Am | Brazil | Duratex | -22.417 | -48.833 | WP | DB | monthly | 54 | 11 | Zanne et al. (2009) | 1/1999-4/2006 | 231.7 (89.7-521.9) | 15.37±0.548 |
| Vieira et al. (2004) | Am | Brazil | FEC | -10.074 | -67.627 | WP | DB | monthly | 313 | 76 | Zanne et al. (2009) | 11/2000-6/2008 | 433.9 (102.7-1388.2) | 36.97±0.558 |
| Zanon and Finger (2010) | Am | Brazil | Flona SFP | -29.417 | -50.404 | WP | DB | monthly | 96 | 1 | Zanne et al. (2009) | 2/2004-6/2006 | 413.1 (235.3-551) | 37.48±0.847 |
| Carvalho (2009) | Am | Brazil | Iaciara | -14.065 | -46.487 | WP | DB | monthly | 171 | 6 | Zanne et al. (2009) | 5/2007-11/2008 | 270.9 (39.3-1815.3) | 18.37±2.965 |
| Rossatto et al. (2009) | Am | Brazil | IBGE | -15.945 | -47.885 | WP | DB | monthly | 116 | 24 | Zanne et al. (2009) | 6/2006-5/2008 | 79.1 (35.7-261.5) | 3.24±0.156 |
| Lisi et al. (2008); Ferreira-Fedele et al. (2004) | Am | Brazil | Ibicatu | -22.783 | -47.717 | WP | DB | monthly | 32 | 5 | Zanne et al. (2009) | 12/1998-5/2006 | 264.2 (109.1-462.1) | 22.44±0.882 |
| Kohler et al. (2008) | Am | Brazil | Irati | -25.374 | -50.575 | WP | DB | 3-monthly | 199 | 20 | Zanne et al. (2009) | 7/2002-6/2008 | 341.6 (100.5-983.1) | 10.52±0.179 |
| de Castro (2014) | Am | Brazil | Itatinga | -23.043 | -48.631 | WP | DB | weekly | 9 | 1 | Zanne et al. (2009) | 11/2012-12/2013 | 52 (45.7-62.9) | 4.02±0.178 |
| Toledo et al. (2012); Paula and Lemos Filho (2001) | Am | Brazil | Lagoa Santa | -19.543 | -43.927 | WP+LP | DB | monthly | 28 | 1 | Toledo et al. (2012) | 10/2009-5/2011 | 322.8 (139.2-711.9) | 9.63±0.991 |
| Grogan and Schulze (2012); Free et al. (2014) | Am | Brazil | Marajoara | -7.833 | -50.267 | WP+LP | DB | monthly | 72 | 3 | Zanne et al. (2009) | 12/1996-11/2001 | 476.3 (137.1-1468.5) | 66.5±1.769 |
| Lisi et al. (2008); Ferreira-Fedele et al. (2004) | Am | Brazil | Porto Ferreira | -21.833 | -47.467 | WP | DB | monthly | 56 | 12 | Zanne et al. (2009) | 12/1998-5/2006 | 314.8 (87.6-883.8) | 20.83±0.893 |
| Kanieski et al. (2012, 2013) | Am | Brazil | REPAR | -25.587 | -49.346 | WP | DB | monthly | 87 | 4 | Zanne et al. (2009) | 7/2009-10/2012 | 190.8 (81.7-325.1) | 5.27±0.168 |



| reference | cont | country | site | Lat | Lon | type | method | time_scale | N_tree | N_sp | wsg | duration | diam | dagh±SE |
|---|---|---|---|---|---|---|---|---|---|---|---|---|---|---|
| Silveira et al.; Vieira et al. (2004) | Am | Brazil | RHF | -9.754 | -67.664 | WP | DB | monthly | 253 | 89 | Zanne et al. (2009) | 1/2005-6/2008 | 326.9 (103.3-1410.4) | 32.83±1.297 |
| Cardoso et al. (2012) | Am | Brazil | Rio Cachoeira | -25.314 | -48.690 | WP | DB | monthly | 121 | 2 | Zanne et al. (2009) | 9/2007-10/2008 | 135.5 (63.1-205.4) | 16.25±0.69 |
| Lisi et al. (2008); Ferreira-Fedele et al. (2004) | Am | Brazil | Santa Genebra | -22.746 | -47.109 | WP | DB | monthly | 22 | 9 | Zanne et al. (2009) | 9/2000-5/2006 | 260.5 (99-554.1) | 11.5±0.75 |
| Lisi et al. (2008); Ferreira-Fedele et al. (2004) | Am | Brazil | SRPQ | -21.667 | -47.500 | WP | DB | monthly | 48 | 8 | Zanne et al. (2009) | 2/2000-12/2006 | 275.4 (199.8-376.9) | 18.66±0.523 |
| Vieira et al. (2004); Nepstad and Moutinho (2013) | Am | Brazil | Tapajos km67 | -2.853 | -54.955 | WP | DB | monthly | 1369 | 263 | Zanne et al. (2009) | 6/1999-3/2006 | 326.2 (99-1997.6) | 18.49±0.35 |
| Figueira (2011); Nepstad and Moutinho (2013) | Am | Brazil | Tapajos km83 | -3.017 | -54.971 | WP+LP | DB | weekly | 734 | 127 | Zanne et al. (2009) | 11/2000-12/2004 | 345.6 (101.3-1135.2) | 32.34±0.412 |
| Lisi et al. (2008); Ferreira-Fedele et al. (2004) | Am | Brazil | Tupi | -22.723 | -47.530 | WP | DB | monthly | 32 | 6 | Zanne et al. (2009) | 12/1998-5/2006 | 224.9 (123.3-483.3) | 16.04±0.824 |
| Chambers et al. (2013) | Am | Brazil | ZF-2 | -2.967 | -60.183 | WP | DB | monthly | 174 | 73 | Zanne et al. (2009) | 7/2000-12/2001 | 222.6 (101.9-644.6) | 5.74±0.245 |
| Mendivelso et al. (2013) | Am | Colombia | Tulua | 4.083 | -76.200 | WP | DB | monthly | 39 | 4 | Mendivelso et al. (2013) | 7/2010-8/2011 | 208.3 (129.4-338.4) | 15.2±0.858 |
| O'Brien et al. (2008); Clark et al. (2010, 2009) | Am | Costa Rica | La Selva | 10.431 | -84.004 | WP+LP | DB | monthly | 205 | 49 | Zanne et al. (2009) | 4/1997-5/2012 | 321.1 (100.3-743.1) | 37.38±0.768 |
| Homeier (2012) | Am | Costa Rica | RBAB | 10.215 | -84.597 | WP | DB | monthly | 403 | 74 | Zanne et al. (2009) | 12/1999-4/2003 | 250.5 (103.3-1000.2) | 5.79±0.101 |
| Homeier et al. (2010, 2012); Roderstein et al. (2005); Brauning et al. (2009) | Am | Ecuador | RBSF | -3.978 | -79.077 | WP+LP | DB,EPD | monthly and 30-min | 694 | 92 | Zanne et al. (2009) | 7/1999-12/2011 | 182.3 (81.8-681.7) | 3.22±0.059 |
| Wagner et al. (2013); Stahl et al. (2010); Bonal et al. (2008) | Am | French Guiana | Paracou | 5.279 | -52.924 | WP+LP | DB | bi-weekly | 256 | 74 | Rutishauser et al. (2010); Stahl et al. (2010); Barakoto et al. (2010) | 4/2007-6/2010 | 337.8 (95.4-1001.6) | 19.21±0.389 |
| Lopez-Ayala et al. (2006) | Am | Mexico | El Palmar | 19.133 | -104.467 | WP | DB | bi-monthly | 23 | 2 | Zanne et al. (2009) | 6/2002-8/2003 | 212.5 (81.3-500.5) | 6.02±0.981 |
| Lopez-Ayala et al. (2006) | Am | Mexico | La Barcinera | 19.150 | -104.425 | WP | DB | bi-monthly | 14 | 1 | Zanne et al. (2009) | 6/2002-8/2003 | 198.3 (96-416.4) | 2.94±0.808 |
| Rowland et al. (2014) | Am | Peru | Tambopata | -12.835 | -69.285 | WP+LP | DB | 3-monthly | 1167 | 287 | Rowland et al. (2014); Zanne et al. (2009) | 10/2005-4/2011 | 221.5 (91.3-1966.3) | 17.37±0.22 |
| Ross et al. (2003) | Am | USA | Big Pine Key | 24.671 | -81.354 | WP | DB | monthly | 15 | 7 | Zanne et al. (2009) | 4/1990-11/1993 | 180.1 (112.8-299.3) | 1.48±0.166 |
| Ross et al. (2003) | Am | USA | Key Largo | 25.267 | -80.324 | WP | DB | monthly | 36 | 15 | Zanne et al. (2009) | 12/1989-11/1993 | 175.4 (103.2-338.4) | 2.52±0.221 |
| Ross et al. (2003) | Am | USA | Lignumvitae Key | 24.903 | -80.698 | WP | DB | monthly | 27 | 11 | Zanne et al. (2009) | 6/1990-11/1993 | 162.3 (99.9-376.6) | 1.45±0.279 |
| Ross et al. (2003) | Am | USA | Sugarloaf Key | 24.625 | -81.543 | WP | DB | monthly | 47 | 12 | Zanne et al. (2009) | 1/1990-11/1993 | 144.5 (101.7-226.6) | 1.35±0.074 |
| Worbes (1999) | Am | Venezuela | RFC | 7.500 | -71.083 | WP | DB | monthly | 25 | 7 | Zanne et al. (2009) | 4/1978-5/1982 | 256.9 (117.2-391.8) | 21.04±1.029 |
| Pelissier and Pascal (2000); Pascal (1984) | As | India | Attapadi | 11.083 | 76.450 | WP+LP | DB | monthly | 101 | 23 | Zanne et al. (2009) | 3/1980-11/1983 | 172.7 (32-1250.9) | 6.21±0.655 |
| Vincent (2012) | As | Indonesia | Muara Bungo | -1.523 | 102.273 | WP | M | monthly | 40 | 3 | Zanne et al. (2009) | 4/2004-5/2006 | 135 (53.3-175.5) | 14.18±0.608 |
| Kho et al. (2013) | As | Malaysia | Lambir | 4.200 | 114.033 | WP+LP | DB | monthly | 1048 | 334 | Kho et al. (2013) | 6/2009-9/2010 | 224.9 (22-1367.1) | 10.2±0.314 |
| Toma (2012) | As | Malaysia | Pasoh | 2.983 | 102.300 | WP | DB | weekly | 195 | 41 | Zanne et al. (2009) | 8/1991-10/1994 | 232.7 (99-688.5) | 14.76±0.506 |
| Ohashi et al. (2009); Bunyavejchewin (1997) | As | Thailand | SERS | 14.500 | 101.933 | WP+LP | DB | monthly | 35 | 7 | Zanne et al. (2009) | 3/2004-10/2006 | 386.7 (161.2-1075.6) | 4.38±0.28 |
| Prior et al. (2004) | Au | Australia | Berry Springs | -12.700 | 131.000 | WP | DB | monthly | 28 | 6 | Zanne et al. (2009) | 11/2000-5/2002 | 122.9 (24.2-287.9) | 2.44±0.328 |
| Drew et al. (2011) | Au | Australia | CSIRO | -12.411 | 130.920 | WP | EPD | daily | 8 | 1 | Cause et al. (1989) | 2/2009-5/2011 | 83 (61-109.7) | 4.78±0.34 |
| Koenig and Griffiths (2012) | Au | Australia | Gunn Point1 | -12.194 | 131.147 | WP | DB | monthly | 6 | 1 | Zanne et al. (2009) | 4/2003-4/2005 | 105.3 (65.4-138.7) | 1.03±0.247 |
| Koenig and Griffiths (2012) | Au | Australia | Gunn Point1B | -12.151 | 131.035 | WP | DB | monthly | 6 | 1 | Zanne et al. (2009) | 4/2003-4/2005 | 205.7 (87.2-324) | 1.82±0.823 |
| Koenig and Griffiths (2012) | Au | Australia | Gunn Point2B | -12.226 | 131.030 | WP | DB | monthly | 6 | 1 | Zanne et al. (2009) | 4/2003-4/2005 | 206.9 (64.7-336.2) | 1.56±1.061 |
| Koenig and Griffiths (2012) | Au | Australia | Gunn Point3 | -12.184 | 131.028 | WP | DB | monthly | 6 | 1 | Zanne et al. (2009) | 4/2003-4/2005 | 107.4 (74.6-141.5) | 1.44±0.297 |
| Brodribb et al. (2013) | Au | Australia | Indian Island | -12.641 | 130.507 | WP | DB | 3-monthly | 20 | 1 | Zanne et al. (2009) | 6/2008-10/2010 | 233.9 (107.7-411.8) | 3.72±0.45 |
| Prior et al. (2004) | Au | Australia | Leanyer | -12.404 | 130.898 | WP | DB | monthly | 12 | 3 | Zanne et al. (2009) | 2/2001-5/2002 | 85 (21.1-189) | 2.46±0.604 |
| Brodribb et al. (2013); Stocker et al. (1995) | Au | Australia | Mt Baldy | -17.269 | 145.423 | WP+LP | DB | 3-monthly | 20 | 1 | Zanne et al. (2009) | 5/2008-8/2010 | 306.3 (171.9-598.4) | 4.37±0.516 |




**Table 3.** Description of the study sites for litterfall measurements; adapted from Chave et al. (2010). For each site, reference of the article, continent, country, full site name and geographical coordinates (long.-lat., in degrees) are reported. The next column reports annual litterfall measurement of wood productivity and litterfall (WP+LP) or only Litterfall (LP), leaf fall (YES) or total litterfall (NO), the number of traps, the trap size, the total area sampled, the mean litterfall productivity in Mg.ha$^{-1}$.year$^{-1}$ and the duration.

| reference | cont | country | site | Lat | Lon | type | typ data | trap nb | trap size | tot size | Mean ± SE | duration |
|---|---|---|---|---|---|---|---|---|---|---|---|---|
| Baker et al. (2003); Owusu-Sekyere et al. (2006) | Af | Ghana | Tinte Bepo | 7.067 | -2.100 | WP+LP | YES | 9 | 1 | 9 | 8.59±1.123 | 1998/2000 |
| Chave et al. (2010) | Am | Brazil | Apiau Roraima | 2.567 | -61.300 | LP | NO | 6 | 1 | 6 | 8.91±0.564 | 1988/1989 |
| Chave et al. (2010) | Am | Brazil | BDFFP Reserve | -2.500 | -60.000 | LP | NO | 18 | 1 | 18 | 6.59±0.675 | 1999/2002 |
| Chave et al. (2010) | Am | Brazil | Capitao Paco Para | -1.733 | -47.150 | LP | NO | 16 | 1 | 16 | 7.97±0.6 | 1979/1980 |
| Castilho et al. (2012) | Am | Brazil | Caracarai | 1.476 | -61.019 | WP+LP | YES | 75 | 0.25 | 18.75 | 5.36±0.19 | 2012/2013 |
| Chave et al. (2010) | Am | Brazil | Caxiuana | -1.785 | -51.466 | LP | YES | 25 | 0.25 | 6.25 | 6.17±0.738 | 2005/2006 |
| Chave et al. (2010) | Am | Brazil | Cuieiras Reserve Manaus | -2.567 | -60.117 | LP | NO | 15 | 0.5 | 7.5 | 8.03±0.564 | 1979/1982 |
| Chave et al. (2010) | Am | Brazil | Cunua-Una Reserve | -2.000 | -54.000 | LP | YES | 45 | 1 | 45 | 6.62±0.799 | 1994/1995 |
| Melgaço (2014); Chave et al. (2010) | Am | Brazil | Ducke | -2.952 | -59.944 | WP+LP | YES | 10 | 0.25 | 2.5 | 3.97±0.197 | 1976/1977 |
| Chave et al. (2010) | Am | Brazil | Jari Para | -1.000 | -52.000 | LP | YES | 100 | 0.25 | 25 | 7.63±0.896 | 2004/2005 |
| Toledo et al. (2012); Paula and Lemos Filho (2001) | Am | Brazil | Lagoa Santa | -19.543 | -43.927 | WP+LP | YES | 20 | 0.2 | 4 | 4.12±0.331 | 1997/1998 |
| Chave et al. (2010) | Am | Brazil | Manaus | -3.133 | -59.867 | LP | NO | 20 | 0.25 | 5 | 7.24±0.607 | 1997/1999 |
| Grogan and Schulze (2012); Free et al. (2014) | Am | Brazil | Marajoara | -7.833 | -50.267 | WP+LP | NO | 50 | 1 | 50 | 3.53±0.416 | 1998/2001 |
| Chave et al. (2010) | Am | Brazil | Mata de Piedade Pernambuco | -7.833 | -34.917 | LP | YES | 10 | 0.25 | 2.5 | 11.05±1.427 | 2003/2004 |
| Chave et al. (2010) | Am | Brazil | Nova Xavantina | -14.685 | -52.335 | LP | YES | 10 | 1 | 10 | 0.45±0.091 | 2002/2003 |
| Chave et al. (2010) | Am | Brazil | Rio Juruena | -10.417 | -58.767 | LP | YES | 16 | 1 | 16 | 5.21±1.514 | 2003/2004 |
| Chave et al. (2010) | Am | Brazil | Sinop | -11.412 | -55.325 | LP | YES | 20 | 1 | 20 | 5.27±1.116 | 2002/2003 |
| Figueira et al. (2011); Nepstad and Moutinho (2013) | Am | Brazil | Tapajos km83 | -3.017 | -54.971 | WP+LP | YES | 30 | 1 | 30 | 5.54±0.533 | 2000/2003 |
| Chave et al. (2010) | Am | Colombia | Amacayacu | -3.717 | -70.300 | LP | YES | 25 | 0.5 | 12.5 | 6±0.31 | 2004/2006 |
| Chave et al. (2010) | Am | Colombia | Chiribiquete | 0.067 | -72.433 | LP | YES | 24 | 0.5 | 12 | 5.62±0.528 | 1999/2002 |
| Chave et al. (2010) | Am | Colombia | Cordillera Central | 4.833 | -75.525 | LP | YES | 30 | 0.25 | 7.5 | 3.36±0.211 | 1986/1987 |
| Chave et al. (2010) | Am | Colombia | Gran Sabana Guayana | 5.117 | -60.933 | LP | NO | 8 | 0.5 | 4 | 5.23±0.449 | 1999/2000 |
| Chave et al. (2010) | Am | Colombia | Zafire | -3.996 | -69.904 | LP | YES | 25 | 0.5 | 12.5 | 5.2±0.383 | 2004/2006 |
| O'Brien et al. (2008); Clark et al. (2010, 2009) | Am | Costa Rica | La Selva | 10.431 | -84.004 | WP+LP | YES | 162 | 0.25 | 40.5 | 6.73±0.314 | 1997/2011 |
| Homeier et al. (2010, 2012); Roderstein et al. (2005); Brauning et al. (2009) | Am | Ecuador | RBSF | -3.978 | -79.077 | WP+LP | YES | 12 | 0.16 | 1.92 | 4.35±0.21 | 2001/2002 |
| Chave et al. (2010) | Am | French Guiana | Nouragues | 4.084 | -52.680 | LP | YES | 40 | 0.5 | 20 | 5.88±0.64 | 2001/2008 |
| Wagner et al. (2013); Stahl et al. (2010); Bonal et al. (2008) | Am | French Guiana | Paracou | 5.279 | -52.924 | WP+LP | YES | 40 | 0.45 | 18 | 4.77±0.311 | 2003/2011 |
| Chave et al. (2010) | Am | French Guiana | Piste de Saint Elie | 5.333 | -53.033 | LP | YES | 60 | 1 | 60 | 5.04±0.608 | 1978/1981 |
| Wieder and J.S. (1995) | Am | Panama | BCI Plateau | 9.154 | -79.846 | LP | NO | 40 | 0.25 | 10 | 12.88±0.941 | 1986/1990 |
| Rowland et al. (2014); Chave et al. (2010) | Am | Peru | Tambopata | -12.835 | -69.285 | WP+LP | YES | 25 | 0.25 | 6.25 | 7.16±0.607 | 2005/2006 |
| Chave et al. (2010) | Am | Venezuela | San Ignacio de Yuruani | 5.000 | -61.017 | LP | NO | 10 | 1 | 10 | 5.23±0.562 | 1990/1991 |
| Pelissier and Pascal (2000); Pascal (1984) | As | India | Attapadi | 11.083 | 76.450 | WP+LP | YES | 100 | 0.5 | 50 | 6.08±0.937 | 1980/1982 |
| Kho et al. (2013) | As | Malaysia | Lambir | 4.200 | 114.033 | WP+LP | YES | 50 | 0.25 | 12.5 | 7.07±0.555 | 2008/2010 |
| Ohashi et al. (2009); Bunyavejchewin (1997) | As | Thailand | SERS | 14.500 | 101.933 | WP+LP | YES | 25 | 1 | 25 | 4.81±0.534 | 1985/1989 |
| Brotribb et al. (2013); Stocker et al. (1995) | Au | Australia | Mt Baldy | -17.269 | 145.423 | WP+LP | YES | 60 | 0.65 | 39 | 5.93±0.48 | 1980/1985 |



**Table 4.** coefficient of the linear model of wood productivity with the precipitation; with all data $m_{WP}$ or after removing the first month of the dry season and wet season (defined respectively as the first month with precipitation $> 100$ mm and the first month with precipitation $< 100$ mm), $m_{WP,-init}$. $^{a}$: confidence intervals.

|  | parameters | 2.5% CI$^{a}$ | 97.5% CI$^{a}$ |
|---|---|---|---|
| $m_{WP}$ | (Intercept) | -0.05 | 0.05 |
|  | precipitation | 0.64 | 0.74 |
| $m_{WP,-init}$ | (Intercept) | -0.08 | 0.02 |
|  | precipitation | 0.61 | 0.72 |



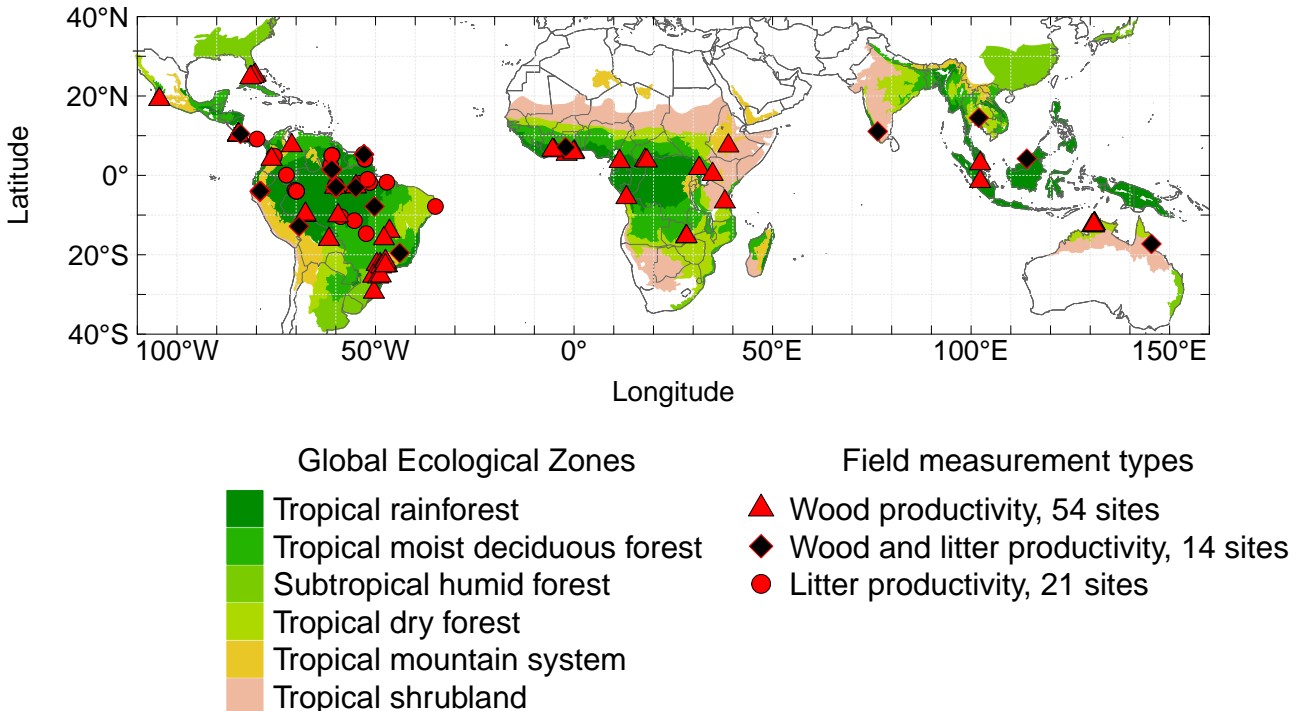

**Figure 1.** Geographical locations of the 89 observation sites with the field measurement types (wood productivity and/or litter productivity) and Global Ecological Zones FAO (2012). Wood productivity is available for 68 sites (54+14), litter productivity for 35 sites (21+14), and EVI and climate for all the 89 studied sites (54+21+14).




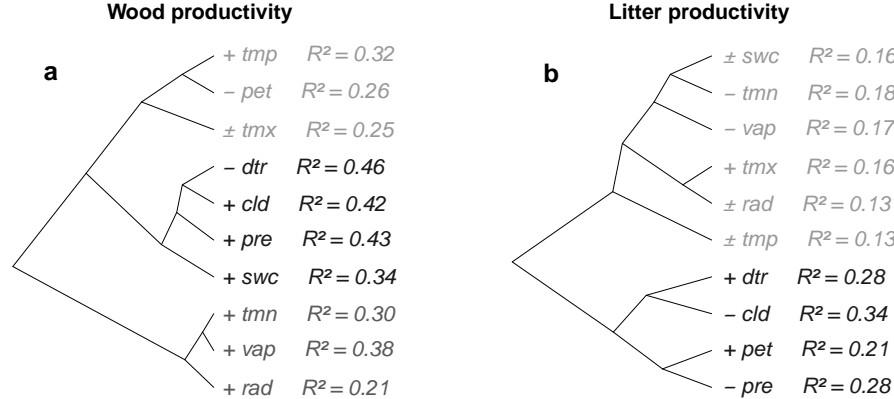

**Figure 2.** Dendrogram of the climate seasonality associations with the seasonality of wood productivity (a) and litterfall (b). The global sign and $R^2$ of the linear relationship between wood and litter productivity and the following climate variable is given. $+$ indicates a positive correlation between the climate variable and wood or litter productivity in all the sites, $-$ a negative correlation in all the sites, while $\pm$ indicates positive correlation for a portion of the sites while negative for the other. Climate variables in the same cluster are highly correlated, that is, they produce the same prediction in terms of values and effects for the same sites. Different shades of grey indicate the relative strength of associations for each cluster with seasonality of wood or litter productivity, black indicates the strongest association. $cld$: cloud cover; $pre$: precipitation; $rad$: solar radiation at the top of the atmosphere; $tmp$, $tmn$ and $tmx$ are respectively the daily mean, minimal and maximal temperatures; $dtr$: temperature amplitude; $vap$: vapour pressure; $pet$: potential evapotranspiration; and $swc$: relative soil water content.





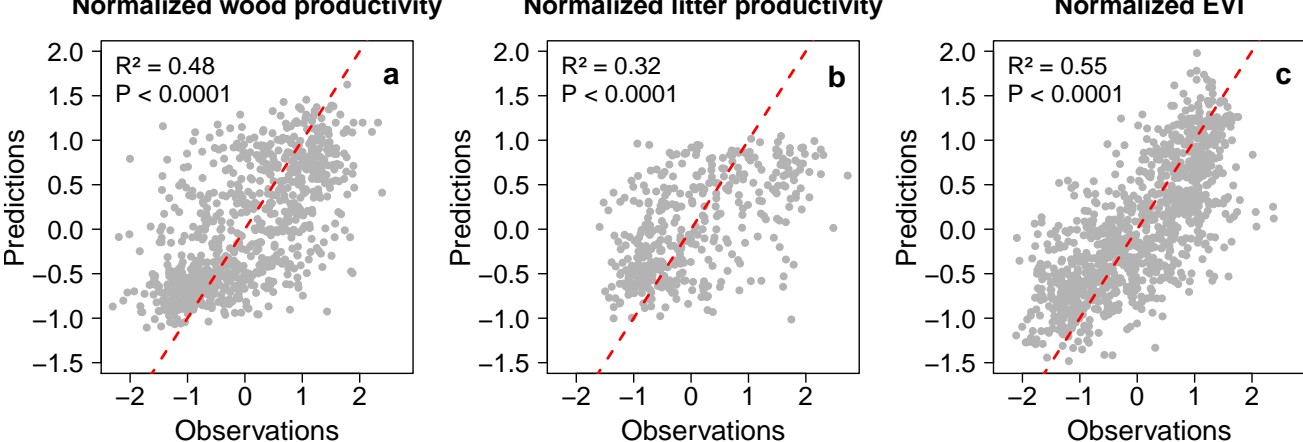

**Figure 3.** Observed versus predicted monthly wood productivity under the model only with precipitation, $m_{WP}$ (a); litterfall productivity under the model only with cloud cover , $m_{lit}$ (b); and EVI the model only with precipitation, maximal temperature and site limitations, $mBIC_{EVI}$ (c). The red dashed line is the identity line y = x. Parameters of the models are given in Table 1.

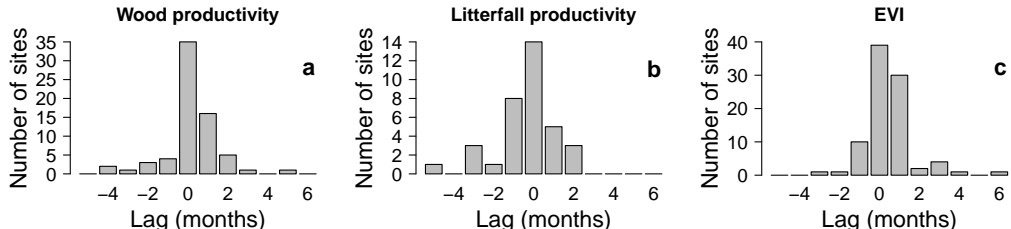

**Figure 4.** Cross correlation between observations and predictions of wood production (a), litterfall (b) and EVI (c) with the linear models parameters (Table 1).



**Figure 5.** Monthly associations of EVI with precipitation (a and b), maximal temperatures (c), and association of $\Delta EVI_{wet-dry}$ with mean annual precipitation (d). In (a) colors represent the value of $\Delta EVI_{wet-dry}$ while in (b), (c) and (d) colors represent $\Delta EVI_{wet-dry}$ grouped by the following classes : water-limited sites ($\Delta EVI_{wet-dry} > 0.0378$), sites with mixed limitations ($\Delta EVI_{wet-dry}$ [-0.0014;0.0378]) and light-limited sites ($\Delta EVI_{wet-dry} < -0.0014$). The dashed lines in (b) and (c) represent the linear relation between the climate variable of the x-axis and EVI obtained with the model mBIC$_{EVI}$ for water-limited sites, sites with mixed limitations and light-limited sites. The dashed lines in (d) represents the best regression model with a breakpoint between $\Delta EVI_{wet-dry}$ and mean annual precipitation.





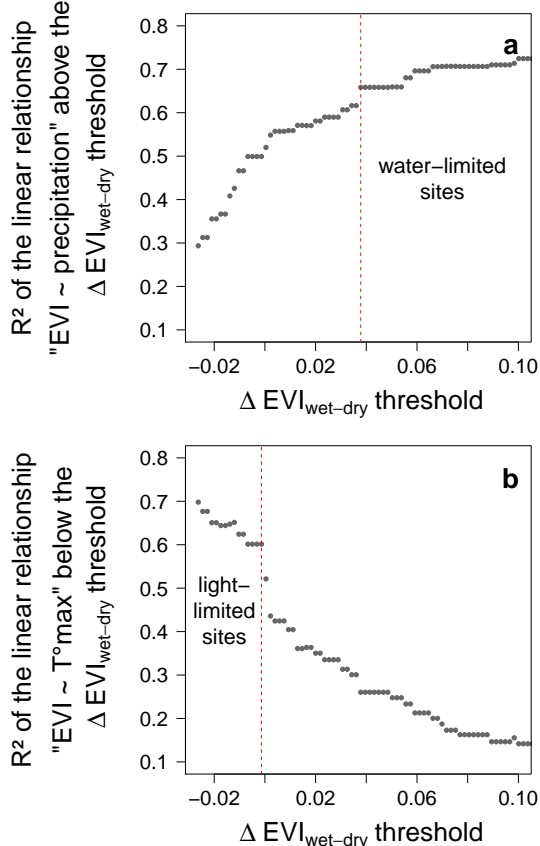

**Figure 6.** Threshold of $\Delta\mathrm{EVI}_{wet-dry}$ used to define 'water-limited' sites (a) and 'light-limited' sites (b). Sites with $\Delta\mathrm{EVI}_{wet-dry}$ between the two thresholds had a mixed influence of the two climate variables and were qualified as 'mixed'. The names of the classes represent the main climate limitations deduced from the climate control on canopy photosynthetic capacity observed in our results. The y-axis represents the $R^2$ values of the linear models normalized EVI as a function of normalized precipitation (a) and as a function of maximal temperature (b), respectively for the sample with $\Delta\mathrm{EVI}_{wet-dry}$ above the threshold (a) and below the threshold (b). Optimal threshold of $\Delta\mathrm{EVI}_{wet-dry}$ for climate variable influence on normalized EVI was defined by a break in the decrease of $R^2$ values, which is represented by red dashed lines.




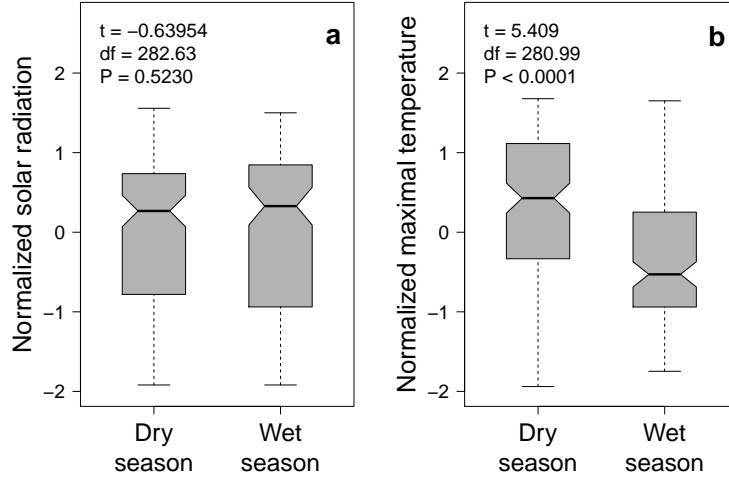

**Figure 7.** Light as an indirect index of solar radiation on the forest floor in light-limited sites. Solar radiation at the top of the atmosphere is not different in dry and wet seasons for these sites, whereas maximal temperature appears to be a good index of the solar insolation at the surface as it integrates both solar radiation and solar interception due to cloud cover. Dry season is defined as months with precipitation < 100 mm.





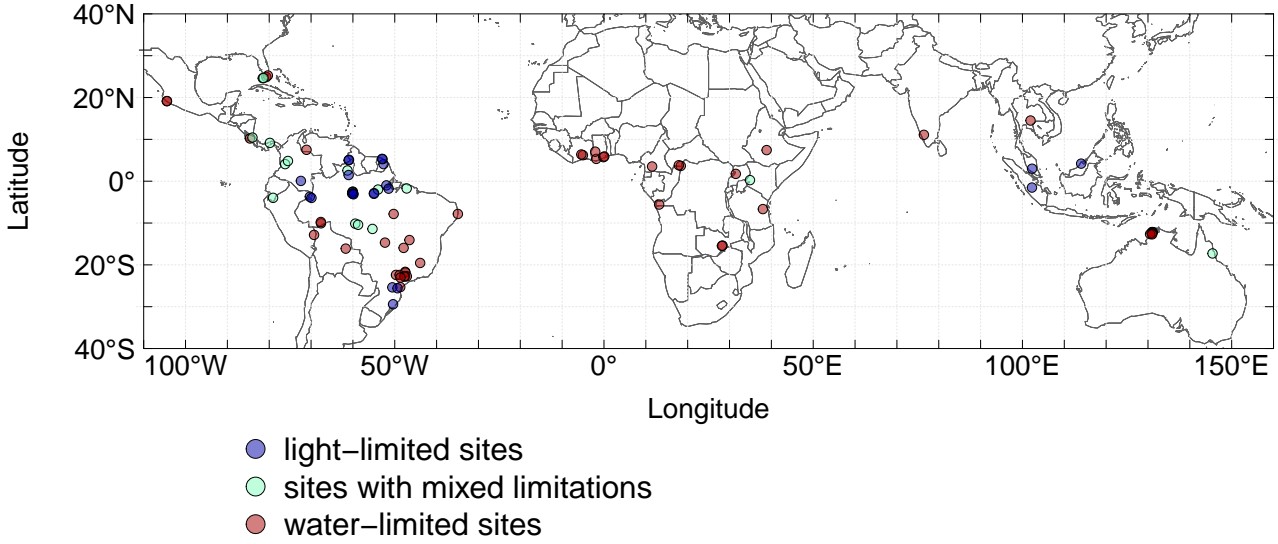

**Figure 8.** Locations and climate limitations of the 89 experimental sites. water-limited sites ($\Delta EVI_{wet-dry} > 0.0378$), sites with mixed limitations ($\Delta EVI_{wet-dry}$ [-0.0014;0.0378]) and light-limited sites ($\Delta EVI_{wet-dry} < -0.0014$), (Fig. 6 for the definition of the thresholds).




**Figure 9.** Observations and predictions of wood productivity and litterfall seasonality in sites where both measurements were available. The outliers in our analysis, Lambir and Caracarai, are not represented. Y-axis have no units as the variables were normalized.




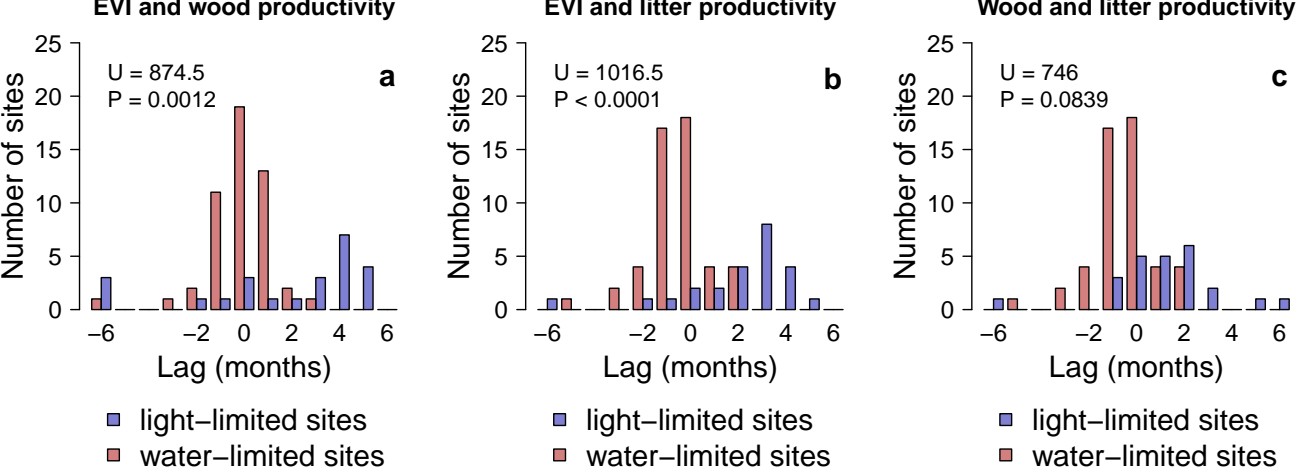

**Figure 10.** Cross-correlation between monthly EVI and wood productivity (a), EVI and litter productivity (b) and wood and litter productivity (c) for water- and light-limited sites. When no observations were available for wood and litter productivity, predictions from the climatic model were used (Table 1). To facilitate graphical representation of cross-correlation (a) is positive, (b) and (c) are negative.





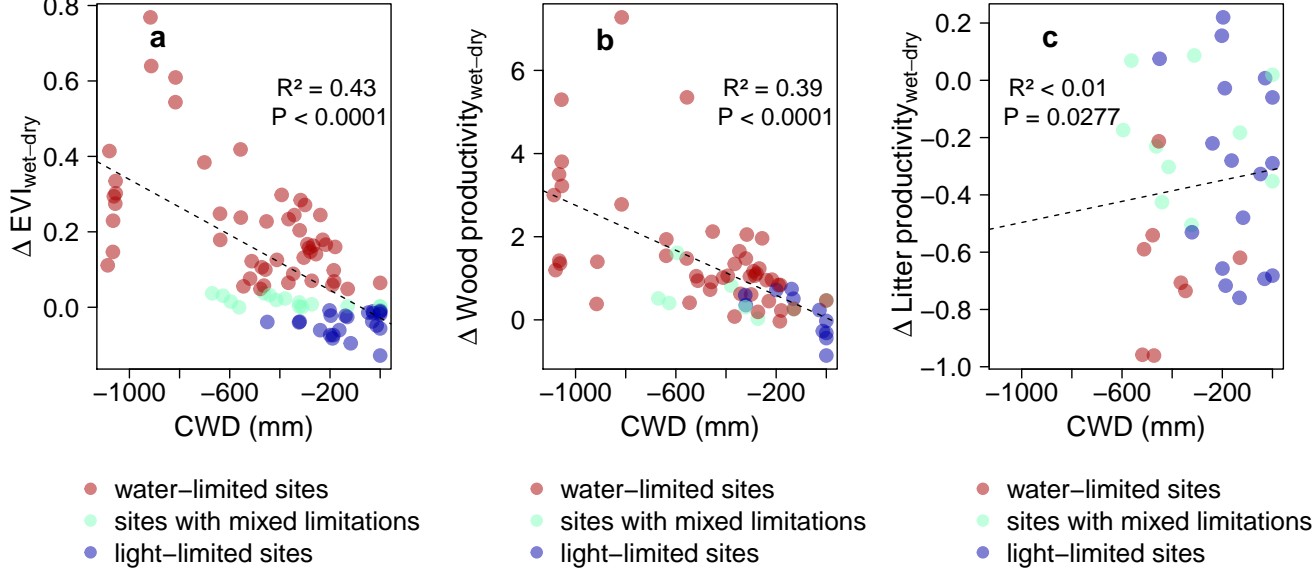

**Figure 11.** Associations between site's $\Delta\mathrm{EVI}_{wet-dry}$ (a), $\Delta\mathrm{Wood}$ productivity$_{wet-dry}$ (b) and $\Delta\mathrm{Litter}$ productivity$_{wet-dry}$ (c) with the environmental variable maximum climatological water deficit ($CWD$). Dashed lines are the regression lines. $\Delta\mathrm{EVI}_{wet-dry}$, $\Delta\mathrm{Wood}$ productivity$_{wet-dry}$ and $\Delta\mathrm{Litter}$ productivity$_{wet-dry}$ indices are the differences of mean of the wet- and dry-season of the variable normalized by the annual mean, where dry season is defined as months with potential evapotranspiration above precipitation (Guan et al., 2015). For the sites where evapotranspiration is never above precipitation, dry season is defined as months with normalized potential evapotranspiration above normalized precipitation.




## SUPPLEMENTARY TABLES

**Table S1.** Number of sites with significant negative (neg), significant positive (pos) or non-significant relationship (no) between the seasonality of wood productivity and each of the climate variables (varclim). Signs $+$ and $-$ indicate the mean sign of the climate variable relationship with the seasonality of wood productivity at lag -1, 0 and +1 month.

| sign (lag -1, 0, +1 month) | varclim | neg | no | pos |
|:---:|:---:|:---:|:---:|:---:|
| $+++$ | pre | 3 | 6 | 59 |
| $+++$ | cld | 2 | 8 | 58 |
| $---$ | dtr | 4 | 9 | 55 |
| $+++$ | swc | 8 | 9 | 51 |
| $+++$ | rad | 2 | 21 | 45 |
| $+++$ | vap | 3 | 21 | 44 |
| $+++$ | tmn | 4 | 21 | 43 |
| $+++$ | tmp | 17 | 15 | 36 |
| $---$ | pet | 13 | 20 | 35 |
| $--+$ | tmx | 20 | 26 | 22 |





**Table S2.** McNemar test of proportion p-values for each of the climate variables used to predict wood productivity. p-value < 0.05 indicates that a different proportion between the two climate variables cannot be rejected.

|     | pre | cld | dtr | vap | tmn | swc | rad | pet | tmp | tmx |
|-----|-----|-----|-----|-----|-----|-----|-----|-----|-----|-----|
| pre | 1.00 | 0.39 | 0.52 | 0.01 | 0.00 | 0.13 | 0.02 | 0.00 | 0.00 | 0.00 |
| cld | 0.39 | 1.00 | 0.54 | 0.02 | 0.01 | 0.20 | 0.02 | 0.00 | 0.00 | 0.00 |
| dtr | 0.52 | 0.54 | 1.00 | 0.01 | 0.00 | 0.53 | 0.02 | 0.00 | 0.00 | 0.00 |
| vap | 0.01 | 0.02 | 0.01 | 1.00 | 0.96 | 0.00 | 0.80 | 0.02 | 0.01 | 0.00 |
| tmn | 0.00 | 0.01 | 0.00 | 0.96 | 1.00 | 0.04 | 0.55 | 0.06 | 0.00 | 0.00 |
| swc | 0.13 | 0.20 | 0.53 | 0.00 | 0.04 | 1.00 | 0.03 | 0.01 | 0.04 | 0.00 |
| rad | 0.02 | 0.02 | 0.02 | 0.80 | 0.55 | 0.03 | 1.00 | 0.04 | 0.00 | 0.00 |
| pet | 0.00 | 0.00 | 0.00 | 0.02 | 0.06 | 0.01 | 0.04 | 1.00 | 0.48 | 0.00 |
| tmp | 0.00 | 0.00 | 0.00 | 0.01 | 0.00 | 0.04 | 0.00 | 0.48 | 1.00 | 0.05 |
| tmx | 0.00 | 0.00 | 0.00 | 0.00 | 0.00 | 0.00 | 0.00 | 0.00 | 0.05 | 1.00 |





**Table S3.** McNemar test of proportion p-values for each of the climate variables used to predict wood productivity for the cluster where *vap* has a positive effect. p-value $< 0.05$ indicates that a different proportion between the two climate variables cannot be rejected. For this subset, *vap* and *pre* are highly correlated ($\rho_{Pearson} = 0.849$, p-value $< 0.001$).

|     | pre  | vap  | tmn  | rad  |
| --- | ---- | ---- | ---- | ---- |
| pre | 1.00 | 0.80 | 0.80 | 0.80 |
| vap | 0.80 | 1.00 | 0.92 | 0.99 |
| tmn | 0.80 | 0.92 | 1.00 | 0.99 |
| rad | 0.80 | 0.99 | 0.99 | 1.00 |





**Table S4.** McNemar test of proportion p-values for each of the climate variables used to predict wood productivity for the cluster where $tmp$ has a positive effect. p-value $< 0.05$ indicates that a different proportion between the two climate variables cannot be rejected. For this subset, tmp and pre are correlated ($\rho_{Pearson} = 0.659$, p-value $< 0.001$).

|      | pre  | tmp  | tmx  | pet  |
| ---- | ---- | ---- | ---- | ---- |
| pre  | 1.00 | 0.80 | 0.02 | 0.00 |
| tmp  | 0.80 | 1.00 | 0.39 | 0.00 |
| tmx  | 0.02 | 0.39 | 1.00 | 0.06 |
| pet  | 0.00 | 0.00 | 0.06 | 1.00 |





**Table S5.** Number of sites with significant negative (neg), significant positive (pos) or non-significant relationship (no) between the seasonality of litter productivity and each of the climate variables (varclim). Signs $+$ and $-$ indicate the mean sign of the climate variable relationship with the seasonality of litter productivity at lag -1, 0 and +1 month.

| sign (lag -1, 0, +1 month) | varclim | neg | no | pos |
|---|---|---|---|---|
| $-\,-\,-$ | cld | 0 | 8 | 27 |
| $+\,+\,+$ | dtr | 1 | 8 | 26 |
| $-\,-\,-$ | pre | 1 | 12 | 22 |
| $+\,+\,+$ | pet | 1 | 14 | 20 |
| $+\,-\,-$ | rad | 4 | 12 | 19 |
| $+\,+\,+$ | tmx | 3 | 13 | 19 |
| $-\,-\,-$ | vap | 3 | 15 | 17 |
| $-\,-\,-$ | tmn | 5 | 13 | 17 |
| $-\,-\,+$ | swc | 5 | 15 | 15 |
| $+\,+\,-$ | tmp | 8 | 15 | 12 |




**Table S6.** McNemar test of proportion p-values for each of the climate variables used to predict litter productivity. p-value < 0.05 indicates that a different proportion between the two climate variables cannot be rejected.

|      | pre  | cld  | dtr  | vap  | tmn  | swc  | rad  | pet  | tmp  | tmx  |
| ---- | ---- | ---- | ---- | ---- | ---- | ---- | ---- | ---- | ---- | ---- |
| pre  | 1.00 | 0.11 | 0.57 | 0.23 | 0.25 | 0.07 | 0.39 | 0.53 | 0.03 | 0.55 |
| cld  | 0.11 | 1.00 | 0.26 | 0.00 | 0.05 | 0.02 | 0.05 | 0.11 | 0.02 | 0.11 |
| dtr  | 0.57 | 0.26 | 1.00 | 0.06 | 0.06 | 0.01 | 0.23 | 0.13 | 0.00 | 0.07 |
| vap  | 0.23 | 0.00 | 0.06 | 1.00 | 0.88 | 0.70 | 0.28 | 0.42 | 0.10 | 0.23 |
| tmn  | 0.25 | 0.05 | 0.06 | 0.88 | 1.00 | 0.78 | 0.88 | 0.43 | 0.76 | 0.92 |
| swc  | 0.07 | 0.02 | 0.01 | 0.70 | 0.78 | 1.00 | 0.69 | 0.26 | 0.39 | 0.51 |
| rad  | 0.39 | 0.05 | 0.23 | 0.28 | 0.88 | 0.69 | 1.00 | 0.54 | 0.43 | 0.94 |
| pet  | 0.53 | 0.11 | 0.13 | 0.42 | 0.43 | 0.26 | 0.54 | 1.00 | 0.01 | 0.53 |
| tmp  | 0.03 | 0.02 | 0.00 | 0.10 | 0.76 | 0.39 | 0.43 | 0.01 | 1.00 | 0.03 |
| tmx  | 0.55 | 0.11 | 0.07 | 0.23 | 0.92 | 0.51 | 0.94 | 0.53 | 0.03 | 1.00 |





**Table S7.** McNemar test of proportion p-values for each of the climate variables used to predict wood productivity for the cluster where tmp has a positive effect. p-value $< 0.05$ indicates that a different proportion between the two climate variables cannot be rejected. For this subset, cld and tmn are correlated ($\rho_{Pearson} = 65.0$, p-value $< 0.001$).

|     | cld  | tmn  | vap  | swc  |
| --- | ---- | ---- | ---- | ---- |
| cld | 1.00 | 0.39 | 0.26 | 0.17 |
| tmn | 0.39 | 1.00 | 0.80 | 0.57 |
| vap | 0.26 | 0.80 | 1.00 | 0.30 |
| swc | 0.17 | 0.57 | 0.30 | 1.00 |



# SUPPLEMENTARY FIGURES

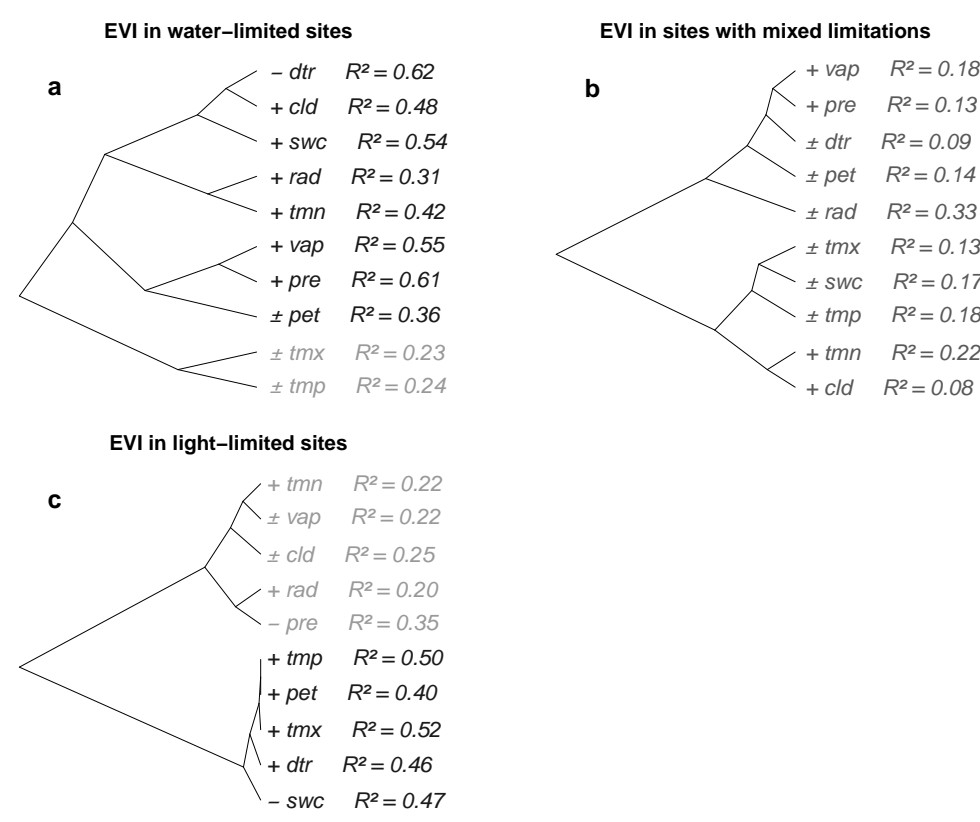

**Figure S1.** Dendrogram of monthly associations of climate variables and EVI for water-limited, mixed and light-limited sites. $+$ indicates a positive correlation between the climate variable and EVI in all the sites of the group (groups: water-limited, mixed or light-limited), $-$ indicates a negative correlation in all the sites of the group, while $\pm$ indicates a positive correlation for a part of the sites of the group while a negative for the other. Climate variables in the same cluster indicates that they are highly correlated, that is, they produce the same prediction in terms of values but also predict the same effect for the same sites. Different shades of grey indicate the relative strength of associations for each cluster with the seasonality of EVI; black indicates the strongest association. $cld$: cloud cover; $pre$: precipitation; $rad$: solar radiation at the top of the atmosphere; $tmp$, $tmn$ and $tmx$ are respectively the daily mean, minimal and maximal temperatures; $dtr$: temperature amplitude; $vap$: vapour pressure; $pet$: potential evapotranspiration; and $swc$: relative soil water content.





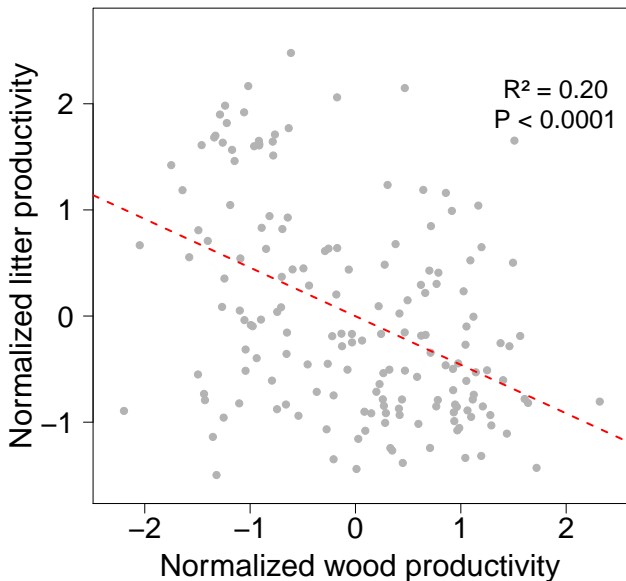

**Figure S2.** Wood productivity versus litter productivity observations. The red dashed line is the linear model between both variables.



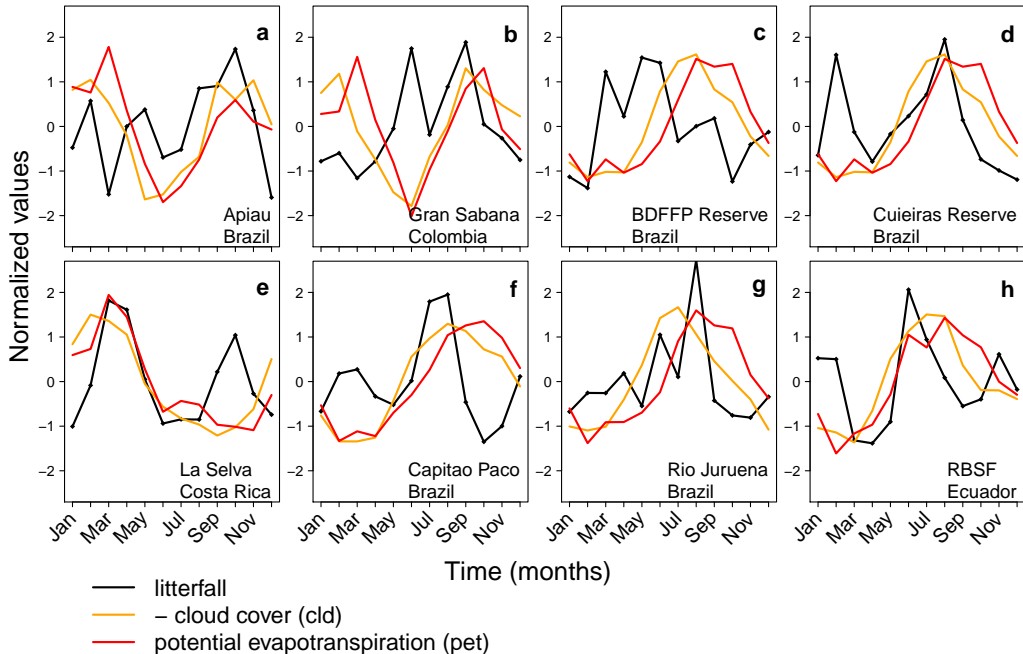

**Figure S3.** Normalized litter productivity, potential evapotranspiration (*pet*) and cloud cover (*cld*) for the sites with no relationship to cloud cover in linear analysis. Cloud cover is multiplied by -1 to facilitate the representation.





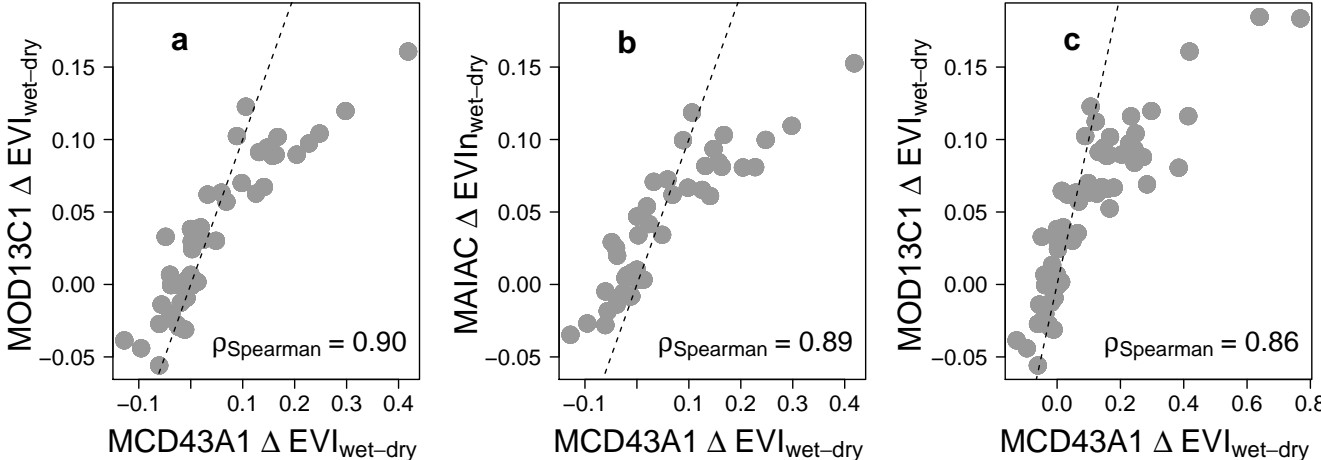

**Figure S4.** Relationships between $\Delta EVI_{wet-dry}$ from MODIS MCD43A1 (this article) and MOD13C1 and MAIAC products for the South American sites (a) and (b), and for all the sites (c) Guan et al. (2015). The climate data used for the computation of $\Delta EVI_{wet-dry}$ from MODIS MCD43A1 (this article) and MOD13C1 and MAIAC products Guan et al. (2015) are independent. The black dashed line is the identity line y = x.