# Peer review of "Climate seasonality limits leaf carbon assimilation and wood productivity in tropical forests."

_Biogeosciences, 2015_

## Referee Comment (RC1) · Anonymous Referee #1 · 21 Jan 2016

This manuscript explores the seasonal correlation of carbon assimilation (estimated using MODIS EVI), above-ground wood productivity and litter productivity. The originality of this work stems in the large dataset compilation made by the authors, which includes data from 89 tropical sites throughout the world. The drivers of the seasonal dynamics of photosynthesis and structural growth has been studied separately in the past, but their joint analysis at large scale is a major novelty of this work. I have no doubt that the focus of this study is of general interest. However, I felt that an additional effort of clarification and justification would benefit the quality of the paper.

General comments

The word "storage" in the title, is in my opinion confusing as it could refer to the non-structural carbohydrate tree compartment. I suggest to consider another word (e.g.,

"above-ground growth").

The authors used statistical analyses that seems overly complex. For example, in order to demonstrate that wood productivity changes are mostly related to seasonal precipitations, the authors estimated simple linear models for each variables, then performed a McNemar test on the resulting contingency tables, before performing a cluster analysis on the table of p-values of the McNemar test. My general feeling is that this complexity might be required (for example in order to deal with collinearity issues among covariates) but must be better justify and better described (see detailed comments).

The logical link among the last three sentences of the first paragraph is not obvious to me. Brienen et al., as well as van der Sleen et al., used annually or multi-annually resolved datasets without explicit reference to the seasonal dynamic of the carbon cycle. I agree that this seasonal dynamic may be key to understanding some of the results reported by the cited authors: e.g., the increasing trend of tree mortality found in Brienen et al. may well be explained by the seasonal dynamic of soil water stress and leaf water potential (e.g., Rowland et al., Nature, 2015), or the seasonal dynamic of non-structural carbon reserve (e.g., Dickman et al., Plant, Cell & Env, 2015). This part of the introduction needs, in my opinion, to be rewritten in order to justify the importance of a better understanding of the seasonal drivers of the carbon cycle. More generally, I felt that this introduction needs to include a broader overview of what is already known about the seasonal dependencies of the carbon cycle to climate and internal factors. Some very interesting results have been obtained (some of them authored by the authors of the present study) regarding the determinism of the seasonal growth of the different tree compartments (wood, leaf, fine root, reserve), as well as the dependencies of photosynthesis in a tropical context. A presentation of these previous works would allow the reader to better understand the novelty and the limits of the present study.

The enhanced vegetation index used by the authors in this study is claimed to be a proxy of "the canopy photosynthetic capacity". Among others, a recent study cited

by the authors (Guan et al., 2015, Nature Geosc.) indeed used EVI as "a proxy for vegetation greenness and photosynthetic potential", which is supported by the strong correlation between EVI and satellite-based chlorophyll fluorescence. But EVI is also known to "include information on forest canopy structure" (Guan et al., 2015, Nature Geosc.). I felt that the complexity of the EVI signal is not emphasized enough in the present paper, as the authors refer to EVI as a "proxy of leaf production", that is, only forest canopy structure. For example, the author have to assume that some big trees at light-limited sites shed leaves because of high evaporative demand, in order to explain the decoupling between EVI and litterfall. It is indeed plausible, but the authors should provide some references to support this assumption, otherwise, they should discuss as well the possibility that – for the range of LAI explored here – EVI changes reflect actually mainly the changes in leaf photosynthesis activity (gC/m2leaf, independently of the total leaf biomass and litterfall). Could you explain why you discarded this latter possibility? I think that your dataset (EVI + litterfall at numerous sites) is very much suited to explore what is actually measured by the EVI.

The authors assumed that a fraction of the 89 studied sites are light-limited, based on reported higher temperatures during dry than during wet season at these sites. Although I agree that light limitation at wet sites is in line with our knowledge of photosynthesis, I felt that this assumption, that is key in this paper, deserve further justification. Indeed, the assumption that temperature actually reflects "solar energy available for the plants" is not supported by references. This is annoying, because temperature is closely link to the water evaporative demand, which is a component of drought. I have the feeling that the correlation between normalized precipitation and normalized EVI at light-limited sites may be significantly negative, which may be in line with the author's assumption (given that precipitation occurs only with a cloud cover). Would you obtain patterns similar to Fig. 5c with a variable more directly related to light availability, such as standardized cloud cover?

Specific comments
P.5. - L.13. "discrepancies" is confusing here, as we do not know which discrepancies the authors are referring to (temporal discrepancies, or discrepancies between biomass and photosynthesis). I suggest using another word, e.g. "patterns".

P.6. - L.11. The tables are not cited in ascending order (Table 2 is cited for the first time before Table 1). The figures are not cited in ascending order either. Please, correct this.

P.6. - L.22:23. "For each tree..." This sentence is unnecessary (and in my opinion, confusing), as the whole process is explained in details in the subsequent sentences. Furthermore, please explain in which cases you deleted the increment, and in which cases you corrected it.

P.7. - L.3. Please, mention here the temporal and spatial resolution of the MODIS product.

P.7. Please, better justify the processing of EVI data (e.g., why did you use a square of 40km? Why did you average pixels surrounding the sites, instead of simply using the values of the pixel sites?).

P.8. - L.5:6. This formulation "normalized by their site's annual mean values and standard deviation" is confusing. Please, give more details about the standardization methodology.

P.9. - L.17. If I am not mistaken, "the predictive model of wood productivity by precipitation" has never been presented before. Consequently, we do not know what the authors are referring to. Please, correct this sentence, and cite the Table 4 in the Results section.

P.12. - L.32. "From the climatic point of view" is not proper English.

P.14. - L.16. I do not see the point to referring to the genetic loci names here. Please, explain in more details how and why this information is relevant.

P.15. - L.5:6. I do not understand this sentence. What is the difference between carbon assimilation and photosynthetic capacity seasonal pattern? Please rephrase.

P.15. - L.7. This last sentence is confusing to me. What is a "direct limitation of canopy photosynthetic capacity" compare to "a reduction of canopy photosynthetic capacity in the dry season". Please rephrase.

Figure 5. Why do the dash lines represent the relationship between climate variable and modelled EVI, rather than observed EVI, in line with the data depicted with dots? The statistic info of the different regression lines should be provided.

Figure 7. Please, rewrite this caption. It does not accurately describe the figure. In my opinion the cross correlation plot, especially Figure 10, do a poor job in illustrating the author's results. For example, I do not understand how Fig. 10a shows that "EVI seasonality is well associated with aboveground wood production for water-limited forests". If I assumed that the Y-axis is actually the "number of sites with a significant correlation", which is not mentioned by the authors, I could evaluate this statement by comparing the number of significant sites against the total number of sites...which is not straightforward. The direct information given by these plots is whether of not the light and water-limited sites have a similar time-lags in their correlation. This information is however not discussed by the authors.
* * *

---

## Referee Comment (RC2) · Anonymous Referee #2 · 14 Feb 2016

**Overall Review**

The manuscript presents a statistical analysis of how seasonality of climate variables is correlated to seasonality of photosynthesis (computed through MODIS EVI product), aboveground wood productivity and litter productivity, the latter data were compiled from a meta-analysis of published literature. The authors found that in wet sites (with approximately precipitation > 2000 mm/yr) photosynthetic capacity and wood productivity are out of phase. In these locations, the EVI seasonality is mostly correlated with maximum temperature interpreted as a proxy for surface radiation, while wood productivity is mostly related to water availability (precipitation). In drier locations, water limitation affects the seasonality of both photosynthetic capacity and wood productivity and their seasonal cycles are temporally correlated. Seasonality of litter productivity correlates less well with climate variables (mostly with cloud cover), and the authors

conclude that endogenous processes as plant phenological strategies could play an important role. The topic and questions addressed by the authors are of general interest, the description of the statistical analysis is sounding and thorough. Even though statistical correlation does not mean causality, the interpretation of the results is based on current knowledge of plant physiological processes and uncertainties are discussed. Most of the presented results are not very novel when compared with what has been already published (e.g., Wagner et al. 2013 Biogeo., Restrepo-Coupe et al 2013 AFM, Guan et al. 2015 Nat. Geo.). However, the authors are aware of this as stated few times in the articles (Page 11, Line 15, Page 12 Line 31, Page 13 Line 16). Even though the results may mostly confirm past studies, the large database assembled by the authors across tropical forests provides additional support and evidence for the pattern of seasonality and relationships with climate in those forests and therefore the article will be likely interesting for many readers. I have just a few minor comments listed below.

**Minor Comments**

*Page 5. Line 20.* I think how it is formulated the third hypothesis "photosynthesis on a global scale is mainly controlled by water limitations" is a bit misleading. I guess with "global scale" the authors just refer to the 89 sites in the tropics, and then although the correlation they found with precipitation is the most significant, this does not exclude other important controls.

*Page 6. Line 19-22 or Page 8 Line 12-21.* The authors may also want to refer to the issue of translating changes in diameter at sub-seasonal scale directly into carbon allocation, it has been recently shown that actual carbon allocation may follow tissue expansion by a considerable amount of time (Cuny et al 2015), or in other words there is a sub-seasonality of wood density in the wood formation period. This should not represent an issue for the present study since the results have been shown to be robust to the exclusion of the first month of wet and dry season (Page 9 Line 12-18) but it is probably worth of mentioning.

[Figure]

*Page 14. Line 25.* I would re-phrase the sentence with more cautious statements, while it is true that photosynthetic capacity and wood productivity correlates with "exogenous variables" there is still a large fraction of unexplained variability. In the presented statistical models, these exogenous variables explain 48

*Page 15. Line 6-8 (also Page 6 Line 5-6 and abstract).* I am not sure, I totally agree with this last sentence. While it is evident than in water limited forests a drier climate will lead to a decline in productivity, in the light-limited forests a drier climate is likely to decrease "cloud-cover" and therefore eventually increase productivity or at least there is no guarantee that water-limitations will become the dominant control and definitely this cannot be inferred from the current analysis.

I would suggest moving Fig. 4 and 10 to the Supp. Material but up to the authors.

Cuny et al. (2015) Woody biomass production lags stem-girth increase by over one month in coniferous forests Nature Plants 1, 15160 doi:10.1038/nplants.2015.160

———————————————

---

## Author Comment (AC1) · 29 Mar 2016

This manuscript explores the seasonal correlation of carbon assimilation (estimated using MODIS EVI), above-ground wood productivity and litter productivity. The originality of this work stems in the large dataset compilation made by the authors, which includes data from 89 tropical sites throughout the world. The drivers of the seasonal dynamics of photosynthesis and structural growth has been studied separately in the past, but their joint analysis at large scale is a major novelty of this work. I have no doubt that the focus of this study is of general interest. However, I felt that an additional effort of clarification and justification would benefit the quality of the paper.

Dear Reviewer#2, thank you very much for your review. In the new version of the paper, we have considered your comments and suggestions, and made several changes in the text. We now believe that the paper is clearer and more accurate.

General comments

The word "storage" in the title, is in my opinion confusing as it could refer to the nonstructural carbohydrate tree compartment. I suggest to consider another word (e.g., "above-ground growth").

FW: the title was changed to "Climate seasonality limits leaf carbon assimilation and wood productivity in tropical forests".

The authors used statistical analyses that seems overly complex. For example, in order to demonstrate that wood productivity changes are mostly related to seasonal precipitations, the authors estimated simple linear models for each variables, then performed a McNemar test on the resulting contingency tables, before performing a cluster analysis on the table of p-values of the McNemar test. My general feeling is that this complexity might be required (for example in order to deal with collinearity issues among covariates) but must be better justify and better described (see detailed comments).

FW: We have added more details on the descriptions and aims of the methodology. The paragraph was changed to "To address the first question 'Are seasonal aboveground wood productivity, litterfall productivity and photosynthetic capacity dependent on climate?', we analyzed with linear models the relationship between our variable of interest (wood productivity, litterfall productivity and photosynthetic capacity) and each climate variable at each site and at t, t-1 month and t+1 month. These lags were chosen to account for between-years variations in the climate seasonality, as we used in our analyses the average climate per site. For a given site, if the wet season have started with one month of delay the year when the tree diameter increment were measured, a lag could exist in the relation of the variable of interest with the monthly averages of climate variables used in linear models. The results were classified for each variable as a count of sites with significantly positive, negative or non-significant results. To enable between-sites comparison, when the overall link was negative, the linear model was finally run with the climate variable multiplied by -1. For a given climate variable, a site with a significant association at only one of the time lags (-1, 0 or 1) was classified as significant. This strategy enables to highlight the potential drivers of our variable of interest, which are the climate variables with a constant relation with the variable of interest in all the sites. Climate variable with no effect, or effect due to a particular correlation with a potential

driver at some sites, will show changes in the sign of the relation with the variable of interest. Then, a McNemar test was run to compare the proportion of our classification (negative, positive or no relationship) between all paired combinations of climate variables accounting for dependence in the data, that is, to compare not only the proportion of positive, negative and no significant effect between two climate variables but also to detect if the sites in each of the classes were similar. In order to summarize all the relations between the climate variables, a table (similar to a correlation table) containing all paired combination p-values of the McNemar test was built. In this table a p-value < 0.05 indicate that a different association between the two climate variables and the variables of interest cannot be rejected. To determine which climate variables explain the same part of variance and to enable interpretation, a cluster analysis was performed on the table of p-values of the McNemar test using Ward distance. Climate variables in the same cluster indicate that they share a similar relation with the variable of interest."

The logical link among the last three sentences of the first paragraph is not obvious to me. Brienen et al., as well as van der Sleen et al., used annually or multi-annually resolved datasets without explicit reference to the seasonal dynamic of the carbon cycle. I agree that this seasonal dynamic may be key to understanding some of the results reported by the cited authors: e.g., the increasing trend of tree mortality found in Brienen et al. may well be explained by the seasonal dynamic of soil water stress and leaf water potential (e.g., Rowland et al., Nature, 2015), or the seasonal dynamic of nonstructural carbon reserve (e.g., Dickman et al., Plant, Cell & Env, 2015). This part of the introduction needs, in my opinion, to be rewritten in order to justify the importance of a better understanding of the seasonal drivers of the carbon cycle.

FW: This part of the introduction have been changed to "Tropical forests have a primary role in the terrestrial carbon (C) cycle, constituting 54% of the total aboveground biomass carbon of Earth's forests (Liu et al., 2015) and accounting for half (1.19 ± 0.41 PgC yr$^{-1}$) of the global carbon sink of established forests (Pan et al., 2011; Baccini et al., 2012). Based on annual or multi-annual measurements of forest wood productivity, changes in carbon dynamics and functioning of the tropical trees have already been observed. While tropical forests have been acting as a long-term, net carbon sink, a declining trend in carbon accumulation has been recently demonstrated for Amazonia (Brienen et al., 2015). Furthermore, a positive change in water-use efficiency of tropical trees due to the $CO_2$ increase over the past 150 years has also been observed (van der Sleen et al., 2015; Bonal et al., 2011). Currently, increasing evidences show that the tropical forests present a seasonality in the assimilation and storage of carbon, associated with climate seasonality (Wu et al., 2016; Doughty et al., 2014; Rowland et al., 2014b, a; Wagner et al., 2014). However, the inherent problems of these studies are that they are one-site or region-based, that renders difficult the disentangling of potential climate drivers due to collinearity between climate variables. Moreover, they sometime focus on a single part of the carbon cycle that may lead to erroneous interpretation on forest productivity due to interactions among the carbon cycle components (Doughty et al., 2014). Understanding the seasonal drivers of the carbon cycle in a pan-tropical context and as well as crossing the maximum information available on carbon storage and assimilation is therefore needed to assess the mechanisms driving changes in forest carbon use and predict tropical forest behavior under future climate changes."

More generally, I felt that this introduction needs to include a broader overview of what is already known about the seasonal dependencies of the carbon cycle to climate and internal factors. Some very interesting results have been obtained (some of them authored by the authors of the present study) regarding the determinism of the seasonal growth of the different tree compartments (wood,

leaf, fine root, reserve), as well as the dependencies of photosynthesis in a tropical context. A presentation of these previous works would allow the reader to better understand the novelty and the limits of the present study.

FW: We have completed this part of the introduction to give more details on the dependencies of the carbon cycle to climate and endogenous factors. The paragraph is now "Despite long-term investigation of changes in forest aboveground biomass stock and carbon fluxes, the direct effect of climate on the seasonal carbon cycle of tropical forests remain unclear. Contrasting results have been reported depending on methods used. Studies show an increase of aboveground biomass gain in the wet season from direct measurement (biological field measurements), or, from indirect measurement, an increase of canopy photosynthetic capacity in the dry season (remote sensing, flux tower network) (Wagner *et al,* 2013}. Several hypotheses have been proposed to explain these patterns: (i) wood productivity, estimated from trunk diameter increment, is mainly controlled by rainfall and water availability and occurs preferentially during the wet season, even if carbon accumulation in the trees could be greater in the dry season than in the wet season, likely reflecting a tradeoff between maximum potential growth rate and hydraulic safety (Rowland *et al,* 2014; Rowland *et al,* 2014b; Wagner *et al,* 2014). Seasonal variation in carbon allocation to the different parts of the plant (crown, roots) also contribute to optimizing resource use and could explain the low synchronicity between wood productivity and carbon accumulation in the trees (Doughty *et al,* 2014; Doughty *et al,* 2015; Rowland *et al,* 2014). (ii) litterfall peak mainly occurs during dry periods as a combination of two potential climate drivers: seasonal changes in daily insolation leading to production of new leaves and synchronous abscission of old leaves, and high evaporative demand and low water availability that both induce leaf shedding in the dry season (Borchert *et al,* 2015; Zhang *et al,* 2014; Wright *et al,* 1990; Chave *et al,* 2010; Myneni *et al,* 2007; Jones *et al,* 2014; Bi *et al,* 2015); and (iii) photosynthesis in these tropical forested regions is mainly controlled by water limitations and is sustained during the dry season above a threshold of 2000 mm of mean annual precipitation (Restrepo-Coupe *et al,* 2013; Guan *et al,* 2015). Water limitation is not the only known control, and other climate variables and internal carbon allocation have been demonstrated to drive photosynthetic capacity in tropical forests such as irradiance, temperature and leaf dynamics. Irradiance is directly and positively linked to plant photosynthetic capacity, carbon uptake and plant growth (Graham *et al,* 2003), while temperatures above 30°C drive a reduction of photosynthetic capacity (Lloyd *et al,* 2008; Doughty *et al,* 2008; Doughty *et al,* 2011). Recently, for non-water-limited forests in Amazonia, Wu et al (2016) showed that the increase in ecosystem photosynthesis during dry period result from the synchronization of new leaf growth and litterfall, shifting canopy composition towards younger more light-use efficient leaves."

The enhanced vegetation index used by the authors in this study is claimed to be a proxy of "the canopy photosynthetic capacity". Among others, a recent study cited by the authors (Guan et al., 2015, Nature Geosc.) indeed used EVI as "a proxy for vegetation greenness and photosynthetic potential", which is supported by the strong correlation between EVI and satellite-based chlorophyll fluorescence. But EVI is also known to "include information on forest canopy structure" (Guan et al., 2015, Nature Geosc.). I felt that the complexity of the EVI signal is not emphasized enough in the present paper, as the authors refer to EVI as a "proxy of leaf production", that is, only forest canopy structure. For example, the author have to assume that some big trees at light-limited sites shed leaves because of high evaporative demand, in order to explain the decoupling between EVI and litterfall. It is indeed plausible, but the authors should provide some references to support this assumption

FW: Recently, Chavana-Bryant et al (2016) have demonstrated that EVI and more generally greenness vegetation index are age-dependent, on a cohort of 1099 leaves from 12 lowland Amazonian canopy trees in southern Peru. Across all trees, EVI and NDVI initially increased with leaf development (from youngest to mature cohorts), and then declined when leaves were at old and senescent stages. Previously, concomitant seasonal increase in leaf flushing and EVI has been reported (Brando et at., 2010, Wagner et al., 2013). These results support our assumption that increase of EVI is mainly linked to maturation of new leaves and that when EVI reached its highest, it seems to represent the moment when leaves are fully mature, the moment of the highest greenness and canopy photosynthetic capacity.

In the introduction we add the sentences: "EVI strongly correlated with chlorophyll content and photosynthetic activity (Huete et al., 2002, 2006), and we used a corrected version of the index to account for sun-angle artifact (Morton et al., 2014; Wagner et al., 2015). While positive correlation of leaf flushing and EVI has already been reported in tropical forests (Brando et at 2010; Wagner et al, 2013; Wu et al, 2016), Chavana-Bryant et al. (2016) have demonstrated that EVI increased with leaf development (from youngest to mature cohorts, and then declined when leaves were at old and senescent stages. Here, we assume that EVI represent the maturation of new leaves and that the highest value of EVI represents the highest greenness and canopy photosynthetic capacity, when leaves are fully mature."

Otherwise, they should discuss as well the possibility that – for the range of LAI explored here – EVI changes reflect actually mainly the changes in leaf photosynthesis activity (gC/m2leaf, independently of the total leaf biomass and litterfall). Could you explain why you discarded this latter possibility? I think that your dataset (EVI + litterfall at numerous sites) is very much suited to explore what is actually measured by the EVI.

FW: As we don't have field measurements of GPP, we choose to assume that EVI is the canopy photosynthetic capacity (or photosynthetic potential) based on Guan et al (2015). Furthermore, even if litterfall production is a proxy of annual leaf production in terms of annual mass balance (Aragão et al 2009), little is known on how litterfall is linked to leaf production at a seasonal scale. Wu et al (2016) indeed showed that litterfall and leaf production occur in the same time in non-water-limited forests of Amazonia (Wu et al, 2016) and they also point out that leaf dynamics give a more accurate prediction of seasonal leaf photosynthesis activity than vegetation indices do, so we prefer to refer to "a proxy of photosynthesis capacity" for EVI. However, even having acknowledged that the results of Wu et al (2016) show that EVI from MAIAC did not reproduce exactly the photosynthetic capacity as well as their dynamic models of leaves in their sites, at the scale of our study, EVI from satellite data represent currently the only manner to have an estimate of photosynthesis capacity.

In the Discussion section we change the sentence "If the increase in EVI is a proxy of leaf production, our result supports..." To "If the increase in EVI is a proxy of leaf maturation, as already observed in a tropical forest of southern Peru (Chavana et al, 2016), our results supports…". And, we add a last sentence to this paragraph: "However, more detailed data on the leaves dynamics would be necessary to confirm these assumptions."

The authors assumed that a fraction of the 89 studied sites are light-limited, based on reported higher temperatures during dry than during wet season at these sites. Although I agree that light limitation at wet sites is in line with our knowledge of photosynthesis, I felt that this assumption, that is key in this paper, deserve further justification. Indeed, the assumption that temperature actually reflects "solar energy available for the plants" is not supported by references. This is annoying, because temperature is closely link to the water evaporative demand, which is a component of

drought. I have the feeling that the correlation between normalized precipitation and normalized EVI at light-limited sites may be significantly negative, which may be in line with the author's assumption (given that precipitation occurs only with a cloud cover). Would you obtain patterns similar to Fig. 5c with a variable more directly related to light availability, such as standardized cloud cover?

FW: Thank you for this comment. Indeed, the assumption that temperature actually reflects "solar energy available for the plants" is not supported by references. Our assumption was that the seasonality of maximal temperature follows the solar cycle, but with a better accuracy of the energy reaching the surface because maximal temperature integrates *de facto* the cloud cover. To test this assumption, we have explored the link between the maximal temperature from the Climate Research Unit data and the incoming radiation at the surface modelled by the CERES. Monthly incident shortwave radiation flux data were obtained from CERES SYN1deg product, version 3A, provided at 1° spatial resolution from March 2000 to Jun 2015. Shortwave (SW) radiation refers to radiant energy with wavelengths in the visible, near-ultraviolet, and near-infrared spectra. The SW radiation flux is produced using MODIS data and geostationary satellite cloud properties (Kato et al., 2011). The SYN1deg product provides datasets calculated for all-sky, clear-sky, pristine (clear-sky without aerosols), and all-sky without aerosol conditions. Here, we used only the product made for all-sky. The results are presented in Fig. 1. This result indicates that the seasonality of maximal temperature is highly correlated with the seasonality of incoming solar radiation at the surface. Seasonal maximal temperature is, hence, a reasonable proxy of the seasonal solar energy available for the plant. The spatial resolution of CERES data is 1° and the time series cover the period between 2000-2015 while the resolution of CRU is 0.5° , encompassing the period between 1901-2012.

[Figure]

$r_{Pearson} = 0.80$
$P < 0.0001$

*Figure 1: Association between normalized maximal temperature from Climate Research Unit and normalized incoming solar radiation at the surface from CERES.*

To answer reviewer #1 question "Would you obtain patterns similar to Fig. 5c with a variable more directly related to light availability, such as standardized cloud cover?", we have reproduce Figure 5c, with maximal temperature Fig. 2 (a) and with incoming solar radiation at the surface from Ceres in Fig 2 (b).

[Figure]

*Figure 2 :*

The pattern in Fig. 2a and 2b are similar. For water-limited sites and sites with mixed limitation, the linear models of EVI as a function of maximal temperature (clear blue dashed lines) or incoming solar radiation (red dashed lines) are not statistically significant. For the light limited sites, the summary of the linear regressions (blue dashed lines) are given in the Table 1. The slopes of the models in light limited sites are both positive but slightly higher for maximal temperature (slope 95% confidence interval of 0.69-0.84 for maximal temperature, and 0.47-0.67 for incoming solar radiation at the surface). The R squared of the models indicate that maximal temperature explains a larger part of the variance of EVI ($R^2$=0.58) than does solar radiation at the surface ($R^2$=0.33).

*Table 1 : Summary of the linear models of normalized EVI as a function or normalized maximal temperature and incoming solar radiation at the surface*

| Linear models | Site limitations | parameter | Estimate | Std. Error | t value | p-value | $R^2$ |
|---|---|---|---|---|---|---|---|
| EVI ~ temperature$_{max}$ | Light limited | (Intercept) | -2.965e-16 | 3.651e-02 | 0.00 | 1 | 0.58 |
| | | tmx | 7.643e-01 | 3.813e-02 | 20.04 | <0.0001 | |
| EVI ~ solar radiation | Light limited | (Intercept) | -1.897e-16 | 4.647e-02 | 0.00 | 1 | 0.33 |
| at the surface | | Sol surface | 5.711e-01 | 4.854e-02 | 11.77 | <0.0001 | |

These additional analyses confirm our interpretation that maximal temperature is a reasonable proxy of solar radiation at the surface. We choose to kept maximal temperature from the CRU in the analysis and not include CERES data of incoming solar radiation, as they represent similar information and because maximal temperature explain a larger part of EVI seasonality ($R^2$=0.58 versus 0.33). However, in Material, we added the sentence "Additionally, we used …..and monthly incoming radiation at the surface (rad$_{surf}$, $W.m^{-2}$) from CERES SYN1deg product computed for all-sky conditions, provided at 1° spatial resolution from 2000 to 2015. Monthly incoming radiation at the surface (shortwave radiation) refers to radiant energy with wavelengths in the visible, near-ultraviolet, and near-infrared spectra and is produced using MODIS data and geostationary satellite cloud properties (Kato et al., 2011).".

Figure 1 included in the responses to reviewer#1 comments, replaced Figure 8 of the article and the legend of Fig. 8 in now: "Association between normalized maximal temperature from Climate Research Unit and normalized incoming solar radiation at the surface from CERES. Monthly incoming solar radiation at the surface (incident shortwave radiation) refers to radiant energy with wavelengths in the visible, near-ultraviolet, and near-infrared spectra and is produced using MODIS data and geostationary satellite cloud properties (Kato et al., 2011).".

In the results, the following sentence was deleted "For these sites, while solar radiation at the top of the atmosphere is not different between the dry and wet seasons, maximal temperature is higher in the dry season, thereby reflecting solar energy available for the plants (Fig. 8)." and replaced by "For all the sites, maximal temperature is highly correlated with incoming solar radiation at the surface (rPearson =0.80, p-value < 0.0001), approximating solar energy available for the plants (Fig. 8)"

Specific comments

P.5. - L.13. "discrepancies" is confusing here, as we do not know which discrepancies the authors are referring to (temporal discrepancies, or discrepancies between biomass and photosynthesis). I suggest using another word, e.g. "patterns".

FW: "discrepancies" was deleted and the word "patterns" is now used.

P.6. - L.11. The tables are not cited in ascending order (Table 2 is cited for the first time before Table 1). The figures are not cited in ascending order either. Please, correct this.

FW: The order of the tables and of the figures had been corrected to be cited in ascending order in the text.

P.6. - L.22:23. "For each tree. . ." This sentence is unnecessary (and in my opinion, confusing), as the whole process is explained in details in the subsequent sentences. Furthermore, please explain in which cases you deleted the increment, and in which cases you corrected it.

FW: The sentence L.22:23. "For each tree. . ." was deleted. The paragraph on the corrections was changed to "If the error was clearly identifiable, such as an abnormal increase (or decrease) in diameter values followed by a large decrease (or increase) of the same amplitude resulting from typo errors, for example 28 whereas 2.8 was expected, the typo error was corrected. When the typo error was not clearly identifiable, the value was corrected with linear approximation with the mean increment of t+1 and t-1. In some cases there was an identifiable increase of diameter values (or decrease), but not followed by a decrease (or an increase) of the same amplitude. This pattern was associated to the repositioning of the dendrometer bands (reported in the source dataset). In this case, the increment was deleted and set to zero and the new time series of cumulative diameter values were computed. As the diameter values are needed to compute biomass, this strategy was used to benefit of the full time series of diameter increment even after solving the error."

P.7. - L.3. Please, mention here the temporal and spatial resolution of the MODIS product.

FW: the sentence "EVI for the 89 experimental sites (Fig. 1) was obtained from the Moderate Resolution Imaging Spectroradiometer (MODIS) MCD43 product collection 5 (4 May 2002 to 30 September 2014)." was changed to "EVI for the 89 experimental sites (Fig. 1) was obtained from the Moderate Resolution Imaging Spectroradiometer (MODIS) MCD43 product collection 5 provided every 16 days at 500m spatial resolution (from 4 May 2002 to 30 September 2014)."

P.7. Please, better justify the processing of EVI data (e.g., why did you use a square of 40km? Why did you average pixels surrounding the sites, instead of simply using the values of the pixel sites?).

FW: In tropical forest regions, few valid MCD43 observations of EVI are available for a pixel, mainly due to obstruction of visible wavelengths by cloud cover and aerosols in the atmosphere. For a given pixel, there is a high probability of missing data in the EVI time series. For resolving this problem, the adopted solution assumed that the forest around the site was homogenous and had the same EVI value of the pixel from the studied site. Then the EVI of the site at a given month was estimated as the mean of all the valid pixels of the surrounding forest. The validity of pixels was defined based on the MCD43 quality index (extracted from MCD43A2 product) from 0 (Good quality) to 3 (All magnitude inversions or 50% or less fill-values). Pixels of quality index 4 (50% or more fill-values) and 255 (Fill-values) were discarded. The pixels representing forests in the 40 km square window were selected based on the land cover map available from the product MCD12Q1 for 2001–2012 at 500 m resolution (Justice et al., 1998); and from the global forest cover loss 2000–2012 and data mask based on Landsat data (Hansen et al., 2013). Only the pixels forested in 2000, without forest losses during the studied period and with tree cover equal or above to the site tree cover were retained.

In this work, we have considered square window of 40km centered on the site to compute the mean EVI to have a reasonable amount of valid pixels to estimate the monthly value of EVI. Using this procedure, only one month at one site had only one value to estimate monthly EVI (Table 1) while 88.8% of the months have 100 or more values to estimate the monthly EVI.

*Table 2: number of valid observations of EVI per months*

| Classes of the number of pixels used to estimate monthly values | Number of months in the classes | Frequency (%) |
|---|---|---|
| (1,2] | 1 | 0.1 |
| (2,5] | 4 | 0.4 |
| (5,10] | 12 | 1.1 |
| (10,50] | 55 | 5.1 |
| (50,100] | 48 | 4.5 |
| (100,>100[ | 948 | 88.8 |

In the text, we have added the sentence "This surface was selected to maximize the quantity of valid pixels to estimate monthly site's EVI, as, due to persistent cloud cover in tropical forest regions, valid observations of EVI are limited, producing incomplete time series of EVI values for a given pixel."

P.8. - L.5:6. This formulation "normalized by their site's annual mean values and standard deviation" is confusing. Please, give more details about the standardization methodology.

FW: The sentence was deleted and changed to "As at some sites, wood productivity or litterfall measurements are older than the EVI measurements (before 2002), and, for recent site measurements, climate data are not yet available (after 2012), all the datasets were monthly averaged by site. Then, in order to remove the site effect on the mean and the variance of the variables and to analyze only seasonality, all the variables were centered and scaled by site. For a given variable of a site, monthly values were subtracted by their annual mean and divided by their annual standard deviation. The obtained normalized variable had a mean of 0 and a variance of 1, but the variation in the variable time-series, that is in our case the seasonality, remained completely unchanged."

P.9. - L.17. If I am not mistaken, "the predictive model of wood productivity by precipitation" has never been presented before. Consequently, we do not know what the authors are referring to.

Please, correct this sentence, and cite the Table 4 (FW: which is now Table 3 with the new table order) in the Results section.

FW: We changed the paragraph to "To test how swelling and shrinking affect our results, we made first a linear model of wood productivity with precipitation as a single predictor with all the data, and then a similar linear model discarding the first month of the wet season (first month with precipitation > 100 mm) and the first month of the dry season (precipitation < 100 mm). Here, we assume that swelling occurs in the first month of the wet season and shrinking occurs in the first month of the dry season, as already observed. The removal of the first month of dry and wet seasons (defined respectively as the first month with precipitation > 100 mm and the first month with precipitation < 100 mm) did not affect the results of the linear model of wood productivity as a function of precipitation, that is, intercepts and slopes are not significantly different in both models (overlaps of the 95\% confidence interval of coefficients and parameters, Table 3)". The parameter of the linear models and the $R^2$ were added to Table 3. As this is a preliminary analysis, we choose to only refer to Table 3 in Methods.

P.12. - L.32. "From the climatic point of view" is not proper English.

FW: "From the climatic point of view" was deleted of the sentence.

P.14. - L.16. I do not see the point to referring to the genetic loci names here. Please, explain in more details how and why this information is relevant.

FW: To clarify, the sentence referring to the genetic loci names was deleted.

P.15. - L.5:6. I do not understand this sentence. What is the difference between carbon assimilation and photosynthetic capacity seasonal pattern? Please rephrase.

FW: Sorry for this, it was an error. The sentence was changed to "Although seasonal carbon allocation to aboveground wood production is driven by water, whether the seasonality of photosynthetic capacity is driven by light or water depends on the limitations of site water availability."

P.15. - L.7. This last sentence is confusing to me. What is a "direct limitation of canopy photosynthetic capacity" compare to "a reduction of canopy photosynthetic capacity in the dry season". Please rephrase.

FW: The last sentence was changed to "In a drier climate, from our results we can make the following assumptions: (i) in water limited forests, the reduction of the wet period duration could lead to a time reduction of favorable conditions for carbon assimilation and allocation. (ii) In current light-limited forests with future precipitation below to the 2000 mm.yr$^{-1}$ threshold, the intensification of the dry period could suppress the canopy photosynthetic capacity increase during this high solar radiation period, reducing carbon assimilation and making these forests shift to water limited forests. However, in light-limited forests with future precipitation above the 2000 mm. yr$^{-1}$ threshold, as cloud cover has been shown to limits net $CO_2$ uptake and growth of tropical forest trees (Graham et al, 2003), it remains uncertain how reduction of cloud cover will affect the productivity."

Figure 5. Why do the dash lines represent the relationship between climate variable and modelled EVI, rather than observed EVI, in line with the data depicted with dots? The statistic info of the different regression lines should be provided.

FW: The dash lines represent the linear relationship between climate variable and observed EVI for water-limited sites, sites with mixed limitations and light-limited sites. We have added a table with

the statistics info of the regression line (Table S8). We have changed the sentence to "The dashed lines in (b) and (c) represent the linear relationship between climate variable and observed EVI for water-limited sites, sites with mixed limitations and light-limited sites. Parameters of the models are given in Supplementary Table S8.

Figure 7. Please, rewrite this caption. It does not accurately describe the figure.

FW: Following the General comment 5 of reviewer 1, the figure was deleted and replace by the figure 1 of the response of reviewer. The title is now "Association between normalized maximal temperature from Climate Research Unit and normalized incoming solar radiation at the surface from CERES. Monthly incoming solar radiation at the surface (incident shortwave radiation) refers to radiant energy with wavelengths in the visible, near-ultraviolet, and near-infrared spectra and is produced using MODIS data and geostationary satellite cloud properties (Kato & al, 2011). The red dashed line is the identity line y = x."

In my opinion the cross correlation plot, especially Figure 10, do a poor job in illustrating the author's results. For example, I do not understand how Fig. 10a shows that "EVI seasonality is well associated with aboveground wood production for water-limited forests". If I assumed that the Y-axis is actually the "number of sites with a significant correlation", which is not mentioned by the authors, I could evaluate this statement by comparing the number of significant sites against the total number of sites. . .which is not straightforward. The direct information given by these plots is whether of not the light and water-limited sites have a similar time-lags in their correlation. This information is however not discussed by the authors.

FW: The sentence "EVI seasonality is well associated with aboveground wood production for water-limited forests" was changed to "In water-limited forests, the seasonality EVI and aboveground wood production are synchronous for the majority of the sites (Fig. 10a), as a consequence of their relationship with precipitation." The title of the figure 10 was changed to "Cross-correlation between monthly EVI and wood productivity (a), EVI and litter productivity (b) and wood and litter productivity (c) for water- and light-limited sites. The x-axis indicates the time-lag to get the maximum correlation between the variables. When no observations were available for wood and litter productivity, predictions from the climatic model were used (Table 4). To facilitate graphical representation, cross-correlation (a) is positive, (b) and (c) are negative. A positive cross-correlation at lag one month indicates a similar seasonal pattern in the time series with a time lag of one month, while a negative cross-correlation at lag one month indicates an opposite seasonal pattern with a time lag of one month. All the water-limited and light-limited sites were represented (respectively 50 and 24 sites) as only 4 water-limited sites in (a) and 3 in (b), and only 2 light-limited sites in (c) have no statistically significant cross-correlation."

Chavana-Bryant, Cecilia; Malhi, Yadvinder; Wu, Jin; Asner, Gregory P.; Anastasiou, Athanasios; Enquist, Brian J.; Cosio Caravasi, Eric G.; Doughty, Christopher E.; Saleska, Scott R.; Martin, Roberta E.; Gerard, France 
[revised manuscript text omitted]

---

## Author Comment (AC2) · 29 Mar 2016

Referee #2

Overall Review

The manuscript presents a statistical analysis of how seasonality of climate variables is correlated to seasonality of photosynthesis (computed through MODIS EVI product), aboveground wood productivity and litter productivity, the latter data were compiled from a meta-analysis of published literature. The authors found that in wet sites (with approximately precipitation > 2000 mm/yr) photosynthetic capacity and wood productivity are out of phase. In these locations, the EVI seasonality is mostly correlated with maximum temperature interpreted as a proxy for surface radiation, while wood productivity is mostly related to water availability (precipitation). In drier locations, water limitation affects the seasonality of both photosynthetic capacity and wood productivity and their seasonal cycles are temporally correlated. Seasonality of litter productivity correlates less well with climate variables (mostly with cloud cover), and the authors C1 BGD Interactive comment Full screen / Esc Printer-friendly version Discussion paper conclude that endogenous processes as plant phenological strategies could play an important role. The topic and questions addressed by the authors are of general interest, the description of the statistical analysis is sounding and thorough. Even though statistical correlation does not mean causality, the interpretation of the results is based on current knowledge of plant physiological processes and uncertainties are discussed. Most of the presented results are not very novel when compared with what has been already published (e.g., Wagner et al. 2013 Biogeo., Restrepo-Coupe et al 2013 AFM, Guan et al. 2015 Nat. Geo.). However, the authors are aware of this as stated few times in the articles (Page 11, Line 15, Page 12 Line 31, Page 13 Line 16). Even though the results may mostly confirm past studies, the large database assembled by the authors across tropical forests provides additional support and evidence for the pattern of seasonality and relationships with climate in those forests and therefore the article will be likely interesting for many readers. I have just a few minor comments listed below.

Minor Comments

Page 5. Line 20. I think how it is formulated the third hypothesis "photosynthesis on a global scale is mainly controlled by water limitations" is a bit misleading. I guess with "global scale" the authors just refer to the 89 sites in the tropics, and then although the correlation they found with precipitation is the most significant, this does not exclude other important controls.

FW: The sentence was changed to "photosynthesis in tropical forested regions is mainly controlled by water limitations". This our general assumption based on the results of other studies, Guan *et al* (2015) which cover pan tropical forested regions and Restrepo-Coupe *et al* (2013) which is based on nine LBA eddy covariance towers in the Brazil flux network (5 are equatorial forest sites). In our results, we confirm this assumption and found another important control for photosynthetic capacity which is solar radiation in sites with precipitation above 2000 mm.yr$^{-1}$, this is a new finding.

Page 6. Line 19-22 or Page 8 Line 12-21. The authors may also want to refer to the issue of translating changes in diameter at sub-seasonal scale directly into carbon allocation, it has been recently shown that actual carbon allocation may follow tissue expansion by a considerable amount of time (Cuny et al 2015), or in other words there is a sub-seasonality of wood density in the wood formation period. This should not represent an issue for the present study since the results have been shown to be

robust to the exclusion of the first month of wet and dry season (Page 9 Line 12-18) but it is probably worth of mentioning.

FW: In Methods, we add the sentence "Recently, Cuny et al. (2015) showed that stem woody biomass production lags behind stem-girth increase by over one month in temperate coniferous, but here we assume that stem-girth increase represent woody biomass production as no such information are yet available for tropical forest trees."

Page 14. Line 25. I would re-phrase the sentence with more cautious statements, while it is true that photosynthetic capacity and wood productivity correlates with "exogenous variables" there is still a large fraction of unexplained variability. In the presented statistical models, these exogenous variables explain 48

FW: The sentence "While photosynthetic capacity and wood productivity appear mostly exogenously driven, litterfall is the result of both exogenous and endogenous processes." was changed to "While photosynthetic capacity and wood productivity appear mostly exogenously driven, litterfall association with climate at seasonal and annual scales suggest both exogenous and endogenous processes." And we add the sentence "It remains that the unexplained variability of photosynthetic capacity and wood productivity seasonality could be link to endogenous drivers but more investigations are needed to demonstrate it".

Page 15. Line 6-8 (also Page 6 Line 5-6 and abstract). I am not sure, I totally agree with this last sentence. While it is evident than in water limited forests a drier climate will lead to a decline in productivity, in the light-limited forests a drier climate is likely to decrease "cloud-cover" and therefore eventually increase productivity or at least there is no guarantee that water-limitations will become the dominant control and definitely this cannot be inferred from the current analysis.

FW: We agree and change the sentences in the conclusion to "In a drier climate, from our results we can make the following assumptions: (i) in water limited forests, the reduction of the wet period duration could lead to a time reduction of favorable conditions for carbon assimilation and allocation. (ii) In current light-limited forests with future precipitation below to the 2000 mm.yr$^{-1}$ threshold, the intensification of the dry period could suppress the canopy photosynthetic capacity increase during this high solar radiation period, reducing carbon assimilation and making these forests shift to water limited forests. However, in light-limited forests with future precipitation above the 2000 mm. yr$^{-1}$ threshold, as cloud cover has been shown to limits net $CO_2$ uptake and growth of tropical forest trees (Graham et al, 2003), it remains uncertain how reduction of cloud cover will affect the productivity."

FW: : In the Abstract, we change the last sentence:"This likely indicates an overall decrease in tropical forest productivity in a drier climate." to "Precipitation first-order control indicates a decrease in tropical forest productivity in a drier climate in water limited forest, and in current light-limited forest with future rainfall < 2000 mm.yr$^{-1}$."

FW: we change the sentence Page 6 Line 5-6:"This likely indicates an overall decrease in tropical forest productivity in a drier climate. To "This likely indicates a decrease in tropical forest productivity in a drier climate in water limited forest, and in current light-limited forest with future rainfall < 2000 mm.yr$^{-1}$."

I would suggest moving Fig. 4 and 10 to the Supp. Material but up to the authors.

FW: Following your comment and the comment of Reviewer #1, we have added information in the legend of the Fig. 4 and the Fig. 10 to clarify the interpretation of the figures.

[revised manuscript text omitted]

---

## Author Response (AR2)

Editorial Review:

FW: Dear Dr. Sönke Zaehle, thank you very much for your review. We have followed your comments and improved the English with the help of a native English speaker.

Unless the paragraph beginning in page 6 L 21 presents results based on previous works (in which case it's missing citations), this paragraph should not occur here (because it introduces the results before the methods). This paragraph reads like a conclusion section to me and should occur either there or in the discussion.

FW: The paragraph was deleted in the introduction, as suggested. It was a summary of our results. All the details of this paragraph are already presented in the discussion and conclusion.

P9 line 19: centered around what? scaled to which range? I assume that the following sentence is meant to explain this, but please remove all doubts.

FW: The sentence was changed to "Then, in order to remove the site effect on the mean and the variance of the variables and to analyze only seasonality, all the variables were centered on zero and scaled to a variance of 1 by site. That is, for a given variable of a site, monthly values were subtracted by their annual mean and divided by their annual standard deviation. The obtained normalized variable had a mean of 0 and a variance of 1, but the time variation in the variable time-series, that is in our case the seasonality, remained completely unchanged."

P 10 L22: Is this relevant? These tree probably did not suffer from water stress, and the growth was primarily thermally controlled. Under these circumstances, why would one expect the measurements to deviate. Or rather - the fact that they don't - what relevance does this have for measurements in seasonally dry trees?

FW: The sentence "Furthermore, a substantial rainfall event occurring after the end of the cambial growth season did not induce xylem initiation or false ring formation" was deleted.

P11 Line 4: Clarify which first question is meant.

FW: it refers to the first of the three research questions presented in the introduction, which are: "(i) Are seasonal aboveground wood productivity, litterfall productivity and photosynthetic capacity dependent on climate? (ii) Does a coherent pan-tropical rhythm exist among these three key components of forest carbon fluxes? (iii) if so, is this rhythm primarily controlled by exogenous (climate) or endogenous (ecosystem) processes?"

P11 Line 8: I don't understand "if the wet season have started with one month of delay the year when the tree diameter increment were measured". Next to "have" -> "has" something seems to be missing here

FW: The sentence was changed to "For example, if the tree diameter increments were measured a year with a wet season initiation delayed of one month in relation to the average year, a lag of one month could exist in the relation of the tree diameter increments and the monthly averages of precipitation used in linear models."

P14 L 22: The "but" is inappropriate here - where is the contradition?

FW: "but" was deleted and the sentence changed to "While this result is not new (Wagner et al, 2014), here we confirm this pattern with a large database of wood production measurements (68 sites).

Please check the many minor language inconsistencies between plural and singular words as well as tenses, such as in the sentence beginning with "However, " in page 5 line 10. There are many more, and this hampers readability of the manuscript.

FW: The manuscript was revised by a native English speaker to correct for these inconsistencies. All the corrections details are also in the manuscript with track changes.

Corrections in the Abstract:

"associate" was changed to "associated"

"to radiation otherwise (light-limited forests); on the other hand," was changed to "to radiation otherwise (light-limited forests). On the other hand,"

"Precipitation first-order control indicates" was changed to "First-order control by precipitation likely indicates"

Corrections in the Introduction:

"Tropical forests have a primary role in the terrestrial carbon (C) cycle, constituting 54% of the total aboveground biomass carbon of Earth's forests (Liu et al., 2015) and accounting for half" was changed to "Tropical forests have a primary role in the terrestrial carbon (C) cycle. They constitute 54% of the total aboveground biomass carbon of Earth's forests (Liu et al., 2015) and account for half"

"increasing evidences show" was changed to "increasing evidence shows"

"region-based, that renders difficult the disentangling of potential climate drivers due" was changed to "region-based, which renders it difficult to disentangle the potential climate drivers due"

"they" was changed to "the studies"

"and as well as crossing" was changed to "by using"

"remain" was changed to "remains"

"contribute" was changed to "contributes"

"litterfall peak mainly occurs during dry periods as a combination of two potential climate drivers" was changed to "litterfall peaks mainly occur during dry periods in response to two potential climate drivers"

"that" was changed to "which"

"controlled by water limitations" was changed to "limited by water"

"dry period result from the synchronization" was changed to "dry periods result from the synchronization"

"represent" was changed to "represents"

Repeated words "canopy photosynthetic capacity" were deleted

"associate" was changed to "associated"

Corrections in the Methods / Datasets:

"We compiled the literature of publications reporting seasonal wood productivity of tropical forests. Seasonal tree growth measurements in 68 pantropical forest sites, 14481 individuals, were obtained from published sources when available or directly from the authors" was changed to "We compiled publications reporting seasonal wood productivity of tropical forests. Seasonal tree growth measurements in 68 pantropical forest sites, representing 14481 individuals, were obtained from published sources or directly from the authors"

"Direct determination for 455 trees" was changed to "Direct determination was available for 455 trees"

"temperate coniferous" was changed to "temperate coniferous forests"

"represent" was changed to "represents"

", increment histogram " was changed to ", the increment histogram"

"For each suspect error, increment" was changed to "For each suspected error, the increment"

"were" was changed to "was"

"typo" was changed to "typographic"

"we used only data with monthly measurements from old-growth forests" was changed to "we used only monthly measurement data from old-growth forests"

"from seasonal pattern" was changed to "from the seasonal pattern"

"As at some sites, wood productivity" was changed to "Because at some sites wood productivity"

"were monthly averaged by site" was changed to "were averaged monthly by site"

Corrections in the Methods / Data analysis:

"Changes in tree circumference with dendrometers" was changed to "Changes in tree circumference measured with dendrometers"

"as they" was changed to "as stems"

"measurements of cambial growth like pinning and microcoring currently represent" was changed to "measurements of cambial growth, such as pinning and microcoring, currently represent"

"have showed" was changed to "have shown"

"initial diameter and significantly" was changed to "initial diameter. This decrease was significantly"

Corrections in the Results:

", but then they are" was changed to ".Rather, they are"

[revised manuscript text omitted]

**SUPPLEMENTARY FIGURES**

[Figure]

**EVI in water–limited sites**

| | |
|---|---|
| − dtr | R² = 0.62 |
| + cld | R² = 0.48 |
| + swc | R² = 0.54 |
| + rad | R² = 0.31 |
| + tmn | R² = 0.42 |
| + vap | R² = 0.55 |
| + pre | R² = 0.61 |
| ± pet | R² = 0.36 |
| ± tmx | R² = 0.23 |
| ± tmp | R² = 0.24 |

**EVI in sites with mixed limitations**

| | |
|---|---|
| + vap | R² = 0.18 |
| + pre | R² = 0.13 |
| ± dtr | R² = 0.09 |
| ± pet | R² = 0.14 |
| ± rad | R² = 0.33 |
| ± tmx | R² = 0.13 |
| ± swc | R² = 0.17 |
| ± tmp | R² = 0.18 |
| + tmn | R² = 0.22 |
| + cld | R² = 0.08 |

**EVI in light–limited sites**

| | |
|---|---|
| + tmn | R² = 0.22 |
| ± vap | R² = 0.22 |
| ± cld | R² = 0.25 |
| + rad | R² = 0.20 |
| − pre | R² = 0.35 |
| + tmp | R² = 0.50 |
| + pet | R² = 0.40 |
| + tmx | R² = 0.52 |
| + dtr | R² = 0.46 |
| − swc | R² = 0.47 |

**Figure S1.** Dendrogram of monthly associations of climate variables and EVI for water-limited, mixed and light-limited sites. $+$ indicates a positive correlation between the climate variable and EVI in all the sites of the group (groups: water-limited, mixed or light-limited), $-$ indicates a negative correlation in all the sites of the group, while $\pm$ indicates a positive correlation for a part of the sites of the group while a negative for the other. Climate variables in the same cluster indicates that they are highly correlated, that is, they produce the same prediction in terms of values but also predict the same effect for the same sites. Different shades of grey indicate the relative strength of associations for each cluster with the seasonality of EVI; black indicates the strongest association. $cld$: cloud cover; $pre$: precipitation; $rad$: solar radiation at the top of the atmosphere; $tmp$, $tmn$ and $tmx$ are respectively the daily mean, minimal and maximal temperatures; $dtr$: temperature amplitude; $vap$: vapour pressure; $pet$: potential evapotranspiration; and $swc$: relative soil water content.

[Figure]

**Figure S2.** Wood productivity versus litter productivity observations. The red dashed line is the linear model between both variables.

[Figure]

**Figure S3.** Normalized litter productivity, potential evapotranspiration ($pet$) and cloud cover ($cld$) for the sites with no relationship to cloud cover in linear analysis. Cloud cover is multiplied by -1 to facilitate the representation.

[Figure]

**Figure S4.** Relationships between $\Delta EVI_{wet-dry}$ from MODIS MCD43A1 (this article) and MOD13C1 and MAIAC products for the South American sites (a) and (b), and for all the sites (c) Guan et al. (2015). The climate data used for the computation of $\Delta EVI_{wet-dry}$ from MODIS MCD43A1 (this article) and MOD13C1 and MAIAC products Guan et al. (2015) are independent. The black dashed line is the identity line y = x.